UPDATE ARTICLE

# Multisensory perceptual and causal inference is largely preserved in medicated post-acute individuals with schizophrenia

**Tim Rohe**[1,2]*, **Klaus Hesse**[1], **Ann-Christine Ehlis**[1,3], **Uta Noppeney**[4]

**1** Department of Psychiatry and Psychotherapy, University of Tübingen, Tübingen, Germany, **2** Institute of Psychology, Friedrich-Alexander-Universität Erlangen-Nürnberg, Erlangen, Germany, **3** Tübingen Center for Mental Health (TüCMH), Tübingen, Germany, **4** Donders Institute for Brain, Cognition and Behaviour, Radboud University, Nijmegen, the Netherlands

* tim.rohe@fau.de

## Abstract

Hallucinations and perceptual abnormalities in psychosis are thought to arise from imbalanced integration of prior information and sensory inputs. We combined psychophysics, Bayesian modeling, and electroencephalography (EEG) to investigate potential changes in perceptual and causal inference in response to audiovisual flash-beep sequences in medicated individuals with schizophrenia who exhibited limited psychotic symptoms. Seventeen participants with schizophrenia and 23 healthy controls reported either the number of flashes or the number of beeps of audiovisual sequences that varied in their audiovisual numeric disparity across trials. Both groups balanced sensory integration and segregation in line with Bayesian causal inference rather than resorting to simpler heuristics. Both also showed comparable weighting of prior information regarding the signals' causal structure, although the schizophrenia group slightly overweighted prior information about the number of flashes or beeps. At the neural level, both groups computed Bayesian causal inference through dynamic encoding of independent estimates of the flash and beep counts, followed by estimates that flexibly combine audiovisual inputs. Our results demonstrate that the core neurocomputational mechanisms for audiovisual perceptual and causal inference in number estimation tasks are largely preserved in our limited sample of medicated post-acute individuals with schizophrenia. Future research should explore whether these findings generalize to unmedicated patients with acute psychotic symptoms.

## Introduction

Hallucinations—percepts in the absence of sources in the external world—are a hallmark of psychotic disorders such as schizophrenia. Individuals with psychosis may for instance hear voices or see people that are not present in their environment. The computational and neural mechanisms that give rise to these perceptual abnormalities remain unclear. Increasing research is guided by the notion that perception relies on probabilistic inference based on 2 distinct sources of information, observers' top-down prior beliefs and bottom-up noisy

**Data Availability Statement:** All raw EEG and behavioral data as well as code for presenting audiovisual stimuli, fitting computational models and analysis of behavioral data are available from a Dryad repository (https://doi.org/10.5061/dryad. hhmgqnkr1) [116].

**Funding:** This work was supported by the Deutsche Forschungsgemeinschaft (https://www. dfg.de/; RO 5587/ 1–1 to TR) and the University of Tübingen (https://uni-tuebingen.de/; Fortüne award 2292–0–0 and 2454–0–0 to TR). Publication costs were covered by the Open Access Publishing Fund of the University of Tübingen. The funders had no role in study design, data collection and analysis, decision to publish, or preparation of the manuscript.

**Competing interests:** UN is an Editorial Board Member of PLOS Biology.

**Abbreviations:** ASD, autism spectrum disorder; BCI, Bayesian causal inference; BDI, Beck's Depression Inventory; BIC, Bayesian information criterion; CDSS, Calgary Depression Scale for Schizophrenia; CMB, crossmodal bias; EEG, electroencephalography; EHI, Edinburgh Handedness Inventory; ERP, event-related potential; GLM, general linear model; HC, healthy control; MWT-B, Mehrfachwahl-Wortschatz-Intelligenztest; PANSS, Positive and Negative Symptom Scale; PCL, Paranoia Checklist; SCZ, schizophrenia; SIFI, sound-induced flash-illusion; SLHS-R, Launay Slade Hallucination Scale; SSP, signal-space projector; SVC, support-vector classifier; SVR, support-vector regression; TMT, Trail Making Test; TR, task relevance; VLMT, verbal learning and memory test.

sensory signals [1–7]. According to Bayesian probability theory, the brain should combine these 2 sources of information weighted according to their relative precisions (i.e., inverse of noise or variance), with a greater weight given to more reliable information. Hallucinations may thus arise from abnormal weighting of prior beliefs and incoming sensory evidence [8,9].

Consistent with this conjecture, overreliance on priors correlated with psychotic symptoms in patients with schizophrenia [10,11] and hallucination-like experiences in hallucination-prone individuals [11–15]. By contrast, direct comparison between patients with schizophrenia and healthy controls did not consistently reveal overreliance on priors but also under-weighting of priors [16–19]. This pattern of results suggests that overreliance on priors may be associated with psychotic symptoms in patients or hallucination-like experiences in healthy controls rather than act as a trait-marker for schizophrenia [12]. However, these diverse findings raise the possibility that psychosis in schizophrenia may either increase or decrease the precision or weight of different types of priors [16,17]. While the weight of priors about simple features (e.g., motion) is thought to decrease in psychosis [18], the weight of priors about semantic or related information may increase [20].

The weighting of various pieces of information becomes even more complex when the brain is confronted with multiple sensory signals that may come from same or different causes. In the face of this causal uncertainty, the brain needs to infer whether 2 sensory signals—say the sound of a whispering voice and the sight of articulatory movements—come from a common source (e.g., a single speaker) and should hence be integrated or else be processed independently (e.g., in case of different speakers). Recent research has shown that hallucinations in psychosis are not only unisensory (e.g., hearing voices), but hallucinations are more often multisensory than previously assumed [21]. Moreover, multisensory hallucinations are associated with greater conviction and more distress for patients [22], especially when signals from different sensory modalities are semantically related or occur simultaneously. This pattern suggests that patients perform causal inference and integrate signals from different sensory modalities into unified hallucinatory percepts. The multisensory nature of hallucinations points towards an additional intriguing mechanism for hallucinations: Patients may be more prone to integrate even signals from different sources and thereby attribute greater "reality status" to them. Critically, because multisensory hallucinations rely on causal inference computations, they cannot be mediated solely by local neural mechanisms in early sensory cortices. Instead, they must involve complex perceptual and causal inference processes along cortical hierarchies.

Models of hierarchical Bayesian causal inference [23–25] account for this causal inference problem in multisensory perception by explicitly modeling the causal structures that could have generated the sensory signals. When signals happen at the same time and space and are semantically (or numerically) congruent, it is likely that they arise from a common source. Hence, in this common-cause case, the brain should fuse the signals weighted in proportion to their relative sensory precisions into 1 single unified perceptual estimate, giving a stronger weight to the more reliable (i.e., less noisy) signal (i.e., fusion estimate). When signals happen at different times or are semantically incongruent, it is likely that they emanate from different sources. In this case of separate sources, the brain should process them independently (i.e., seg-regation estimates for the unisensory signals). Critically, the brain does not a priori know whether signals come from common or independent sources. Instead, it needs to infer the underlying causal structure from noisy statistical correspondence cues such as signals happening at the same time or space, being numerically or semantically congruent [26–29]. To account for observers' uncertainty about the signals' causal structure, the brain should read out a final perceptual estimate by combining the fusion (i.e., common source) and segregation (i.e., separate sources) estimates weighted by the posterior probabilities of each causal structure (i.e., common or independent sources). This decision strategy is referred to as model averaging

(for other decisional functions, see [30]). Bayesian causal inference thereby enables a graceful transition from integration for (near-) congruent auditory and visual signals to segregation for incongruent signals. Behaviorally, accumulating research has shown that human observers arbitrate between sensory integration and segregation consistent with models of Bayesian causal inference [23,27,28,31–36]. At small discrepancies, they combine signals into one coherent percept which leads to prominent crossmodal biases. At larger disparities, these multisensory interactions and crossmodal biases are attenuated. At the neural level, recent research has shown that the brain accomplishes Bayesian causal inference by dynamically encoding the segregation, fusion, and the final perceptual estimate that accounts for the brain's causal uncertainty along cortical pathways [27,33,36–39]. Multisensory perceptual inference is thus governed by 2 sorts of priors, observers' perceptual priors about environmental properties (e.g., the number of signals) as well as a causal prior about whether signals come from common or independent sources, which represents observers' a priori tendency to bind sensory signals. While the former priors influence observer's perceptual estimates directly, the latter does so indirectly by modulating the strength of cross-sensory interactions.

The intricacies of multisensory perception may explain the inconsistent findings regarding multisensory abnormalities in psychosis [40–43] (for a review, see [44]). For example, the rate at which participants with schizophrenia experience the McGurk- or sound-induced flash illusions has been shown to be lower [40,43,45], equal [46], or even higher [41] compared to healthy controls. These inconsistencies may arise from the complex interplay of an individual's auditory and visual precisions, perceptual and causal priors, and decisional strategies, which may all be altered in psychosis. Bayesian modeling and formal model comparison moves beyond previous descriptive approaches by allowing us to dissociate these distinct computational ingredients [47–49]. For example, psychosis may alter how the brain weights prior knowledge and different pieces of sensory information: The brain may over-rely on prior information or even assign a greater weight to information from a specific sensory modality. Psychosis may also increase observers' tendency to bind signals across different senses as quantified by the causal prior. This could facilitate the emergence of percepts that misbind incongruent sensory signals. Finally, psychosis may alter how observers read out their perceptual estimates from complex (e.g., bimodal) posterior distributions that typically arise through Bayesian causal inference [28]. For example, instead of model averaging that is predominantly observed in healthy individuals [27,28,36], patients may apply suboptimal or heuristic strategies that do not optimally take the causal structure of the signals into account [30,31]. This brief overview highlights the powerful insights that may be obtained by combining Bayesian causal inference models with behavioral and neuroimaging data acquired in more complex and challenging multisensory environments.

This psychophysics-EEG study investigated whether schizophrenia alters the computational and/or neural mechanisms of multisensory perception in a sound-induced flash illusion paradigm. In an inter-sensory selective attention task, patients with schizophrenia and age-matched healthy controls were presented with sequences of a varying number of flashes and beeps. We first assessed whether schizophrenia altered the computations of how observers combined auditory and visual signals into number estimates by comparing normative and approximate Bayesian causal inference (BCI) models. Next, we combined BCI modeling with multivariate EEG analyses to unravel the underlying neural mechanisms. Our results suggest that the core computational mechanisms underlying perceptual and caudal inference are largely preserved in our limited medicated schizophrenia cohort compared to healthy participants at the behavioral, computational, and neural level.

## Results

### Experimental design and analysis

Twenty-three healthy controls (HCs) and 17 observers with schizophrenia (SCZ) were included in the behavioral and EEG analyses. HC were matched to SCZ individuals in sex, age, and education. Neuropsychological tests revealed comparable performances in attentional and executive functions across the 2 groups (Table 1). Yet, HC had better memory recall and pre-morbid crystallized intelligence.

In a sound-induced flash-illusion (SIFI) paradigm, we presented HC and SCZ with flash-beep sequences and their unisensory counterparts. Across trials, the number of beeps and flashes varied independently according to a 4 (1 to 4 flashes) × 4 (1 to 4 beeps) factorial design (Fig 1A and 1B). Thereby, the paradigm yielded numerically congruent or incongruent flash-beep sequences at 4 levels of audiovisual numeric disparity. In an inter-sensory selective attention task, observers reported either the number of beeps or flashes. The manipulation of numeric disparity across several levels enabled us to characterize how observers gracefully transitioned from integration to segregation as a key feature of Bayesian causal inference [27,35,50,51].

### Behavior—scalar variability, response accuracy, and crossmodal bias

Fig 2A shows the reported flash and beep counts in SCZ and HC as a function of the true flash and beep numbers. Both SCZ and HC progressively underestimated the increasing numbers of

**Table 1. Demographic, psychopathological and neuropsychological data (across-participants mean ± STD) from HC and participants with SCZ.**

| | HC | SCZ | $t_{38}$ | p | $BF_{10}$ |
|---|---|---|---|---|---|
| *n* | 23 | 17 | - | - | - |
| Age (years) | 35.96 ± 2.50 | 33.12 ± 1.99 | 0.84 | 0.414 | 0.412 |
| Sex (% w) | 43.5 | 17.6 | 2.97(χ2) | 0.085 | - |
| Education (years) | 16.65 ± 0.63 | 15.44 ± 0.89 | 1.14 | 0.269 | 0.520 |
| PANSS Pos. Sympt. (Σ items) | - | 13.59 ± 0.97 | - | - | - |
| PANSS Neg. Sympt. (Σ items) | - | 17.29 ± 1.75 | - | - | - |
| PANSS Gen | - | 28.88 ± 1.71 | - | - | - |
| LSHS-R | - | 13.82 ± 8.85 | - | - | - |
| PCL | - | 38.69 ± 17.94 | - | - | - |
| CPZ equiv | - | 572.59 ± 62.95 | - | - | - |
| VLMT free recall (Σ trial 1–5) | 57.83 ± 1.65 | 49.41 ± 2.99 | 2.63 | 0.009 | 4.230 |
| VLMT delayed recall (trial 7) | 12.61 ± 0.56 | 10.47 ± 0.64 | 2.49 | 0.018 | 3.287 |
| VLMT recognition (Hits-FA) | 13.87 ± 0.32 | 12.91 ± 0.54 | 1.61 | 0.104 | 0.853 |
| TMT A (sec) | 28.60 ± 1.98 | 25.70 ± 2.99 | 0.84 | 0.406 | 0.413 |
| TMT B (sec) | 59.89 ± 3.09 | 67.75 ± 10.72 | −0.80 | 0.462 | 0.401 |
| MWT-B (raw value) | 30.78 ± 0.67 | 27.57 ± 1.32 | 2.34 | 0.019 | 2.523 |
| Stroop effect (sec) | 28.93 ± 2.34 | 33.61 ± 3.49 | −1.16 | 0.258 | 0.527 |
| EHI (Laterality index) | 58.19 ± 10.65 | 69.42 ± 8.23 | −0.79 | 0.439 | 0.398 |
| BDI (Σ items) | 2.09 ± 0.62 | 13.47 ± 1.68 | −7.05 | <0.001 | >100 |
| CDSS (∅ items) | - | 0.33±0.08 | - | - | - |

Note: PANSS: Positive and Negative Symptom Scale (with 3 subscales: Positive Symptoms/Pos. Sympt., Negative Symptoms/Neg. Sympt. and General Psychopathology/PANSS Gen); LSHS-R: Launey-Slade hallucination scale—revised; PCL: Paranoia check list; CPZ equiv: Chlorpromazine equivalents (mg); VLMT: Verbal learning and memory test; TMT: Trail Making Test; MWT-B: Mehrfachwahl-Wortschatz-Intelligenztest (Multipe word choice intelligent test); Stroop effect: time for naming incongruent color words—time for naming color plates; EHI: Edinburgh Handedness Inventory; BDI: Beck's Depression Inventory; CDSS: Calgary Depression Scale for Schizophrenia with 9 subscales.

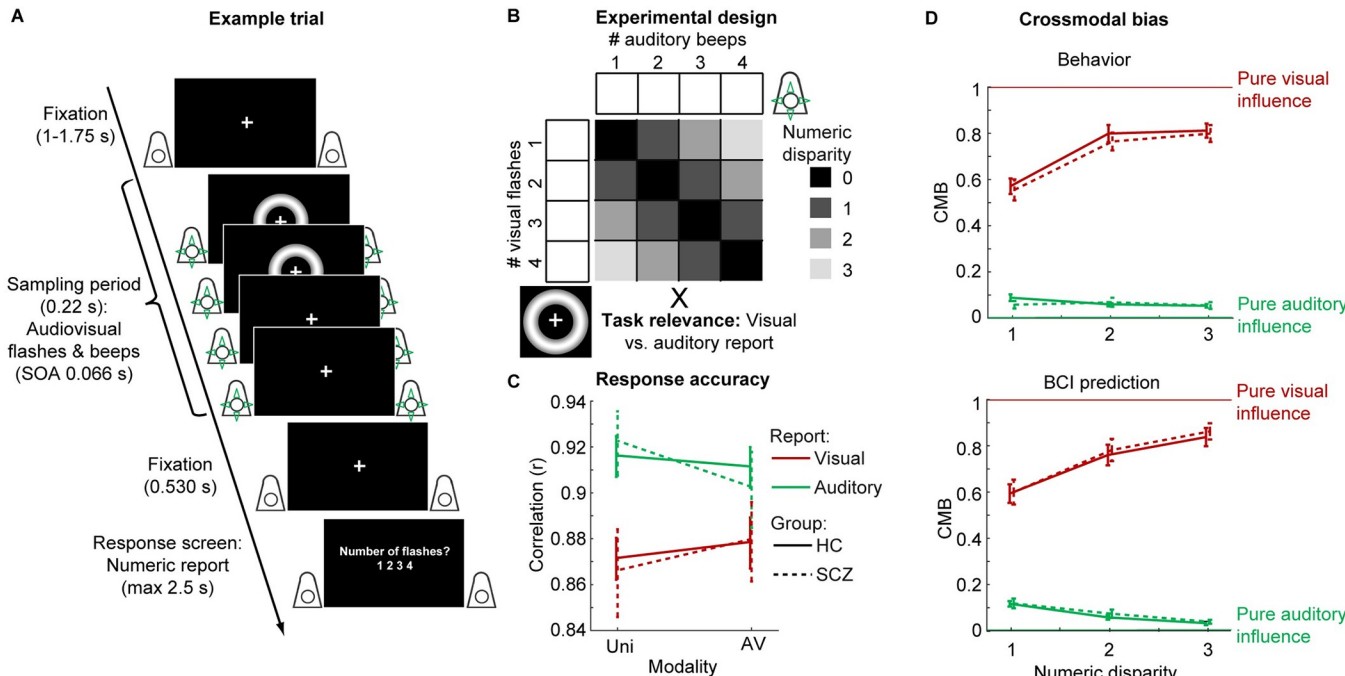

**Fig 1. Example trial, experimental design, and behavioral data.** **(A)** Example trial of the flash-beep paradigm (e.g., 2 flashes and 4 beeps are shown) in which participants either report the number of flashes or beeps. **(B)** The experimental design factorially manipulated the number of beeps (i.e., 1 to 4), number of flashes (i.e., 1 to 4) and the task relevance of the sensory modality (report number of visual flashes vs. auditory beeps). We reorganized these conditions into a 2 (task relevance: auditory vs. visual report) × 2 (numeric disparity: high vs. low) factorial design for the GLM analyses of the audiovisual crossmodal bias. **(C)** Response accuracy (across-participants mean ± SEM; $n = 40$) was computed as correlation between experimentally defined task-relevant and reported signal number. Response accuracy is shown as a function of modality (audiovisual congruent conditions vs. unisensory visual and auditory conditions), task relevance (auditory vs. visual report), and group (HC vs. SCZ). **(D)** The audiovisual CMB (across-participants mean ± SEM; $n = 40$) is shown as a function of numeric disparity (1, 2, or 3), task relevance (auditory vs. visual report) and group (HC vs. SCZ). CMB was computed from participants' behavior (upper panel) and from the prediction of the individually fitted BCI model (lower panel; i.e., model averaging with increasing sensory variances). CMB = 1 for purely visual and CMB = 0 for purely auditory influence. Source data is provided in S1 Data. BCI, Bayesian causal inference; CMB, crossmodal bias; GLM, general linear model; HC, healthy control; SCZ, schizophrenia.

flashes and beeps (S1 Text, S1 Fig, and S1 Table). This logarithmic compression and the increasing variances for greater number of flashes/beeps was observed similarly in both groups. It is in line with the known scalar variability of numerosity estimates [52,53]. Fig 2 also indicated that observers' reported flash (resp. beep) counts were biased towards the concurrent incongruent beep (resp. flash) number in the ignored sensory modality. We formally compared SCZ and HC in their ability to selectively estimate either the number of flashes or beeps (see Fig 1C and Table 2) using a 2 × 2 × 2 mixed-model ANOVA, with factors group (SCZ vs. HC), stimulus modality (unisensory visual/auditory vs. audiovisual congruent), and task relevance (TR) (auditory vs. visual report) on response accuracies (n.b. no incongruent conditions were included in this ANOVA). No significant group differences or interactions with group were observed. Instead, Bayes factors provided substantial evidence for comparable accuracies of the flash and beep counts in HC and SCZ (i.e., $BF_{incl} < 1/3$, Table 2; see S2 Fig for comparable response time results).

As expected [54], we observed a significant main effect of task-relevance and a task-relevance × modality interaction on response accuracies. Overall, observers estimated the number of beeps more accurately than the number of flashes. They also estimated the number of beeps better when presented alone than together with flashes, while the reverse was true for observers' flash counts. Consistent with extensive previous research showing a greater

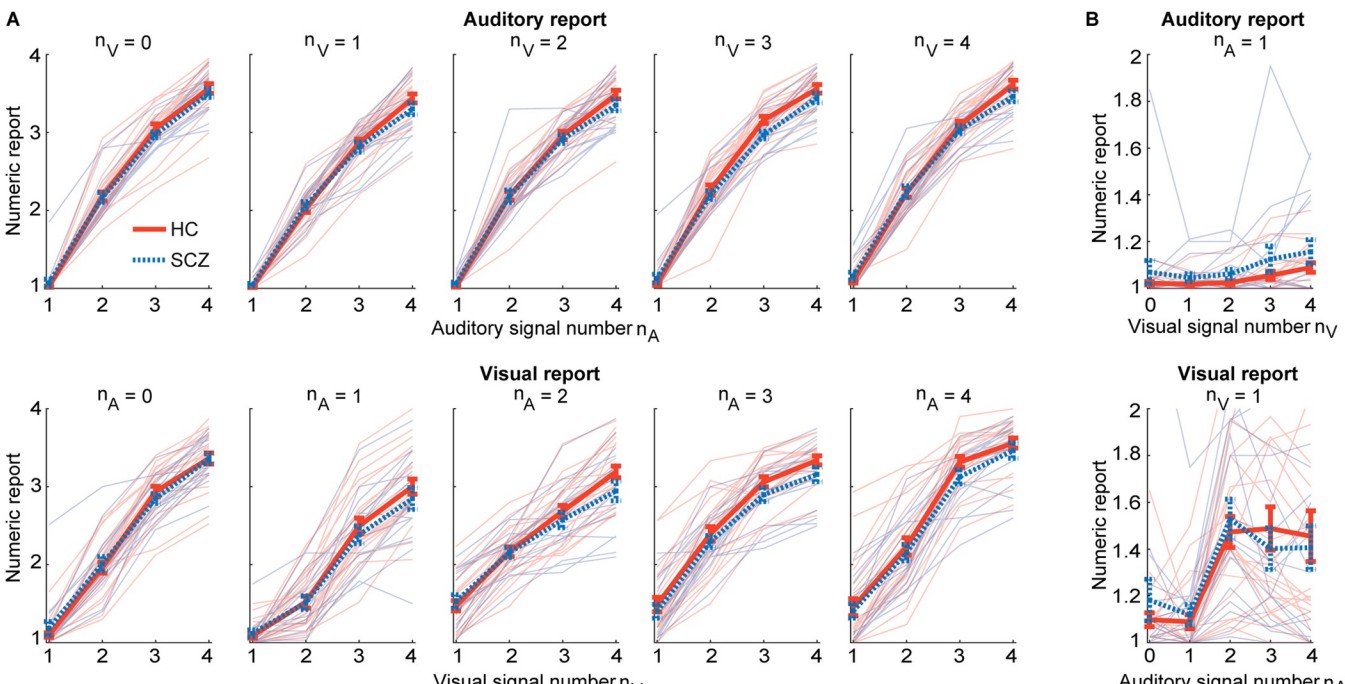

**Fig 2. Distributions of numeric reports (across-participants mean ± SEM, *n* = 40) for HC and SCZ patients. (A)** Upper panel: The auditory numeric reports plotted as a function of auditory signal number $n_A$, separately for different visual signal numbers $n_V$. Lower panel: The visual numeric reports plotted as a function of visual signal number $n_V$, separately for different auditory signal numbers $n_A$. **(B)** Auditory reports for a single beep as a function of visual signal number (upper panel) and visual reports for a single flash as a function of auditory signal number (lower panel). Source data is provided in S2 Data. HC, healthy control; SCZ, schizophrenia.

**Table 2. Main and interaction effects of TR, modality and group on response accuracy and effects of task relevance, disparity, and group on the crossmodal bias from classical and Bayesian mixed-model ANOVAs.**

|  | F | df1, df2 | *p* | part. $\eta^2$ | BF$_{incl}$ |
|---|---|---|---|---|---|
| **Response accuracy** |  |  |  |  |  |
| TR | 49.548 | 1,38 | <0.001 | 0.566 | >100 |
| Mod | 0.590 | 1,38 | 0.447 | 0.015 | 0.602 |
| Group | 0.006 | 1,38 | 0.939 | <0.001 | 0.233 |
| TR×Mod | 6.693 | 1,38 | 0.014 | 0.150 | 2.452 |
| TR×Group | 0.004 | 1,38 | 0.950 | <0.001 | 0.352 |
| Mod×Group | 0.625 | 1,38 | 0.434 | 0.016 | 0.186 |
| TR×Mod×Group | 1.691 | 1,38 | 0.201 | 0.043 | 0.145 |
| **Crossmodal bias** |  |  |  |  |  |
| TR | 633.673 | 1,38 | <0.001 | 0.943 | >100 |
| Disp | 92.312 | 1.7,65.7 | <0.001 | 0.708 | >100 |
| Group | 0.312 | 1,38 | 0.579 | 0.008 | 0.189 |
| TR×Disp | 90.781 | 1.6,61.4 | <0.001 | 0.705 | >100 |
| TR×Group | 0.075 | 1,38 | 0.786 | 0.002 | 0.068 |
| Disp×Group | 0.456 | 1.7,65.7 | 0.608 | 0.012 | 0.146 |
| TR×Disp×Group | 0.914 | 1.6,61.4 | 0.388 | 0.023 | 0.049 |

Note: TR = task relevance: auditory vs. visual report; Mod = modality: audiovisual congruent vs. unisensory; Group: HC vs. SCZ; Disp = absolute numeric disparity (1 vs. 2 vs. 3). Effects of the classical mixed-model ANOVA were Greenhouse–Geisser corrected if sphericity was violated.

temporal precision for the auditory than the visual sense [55], our results confirm that observers obtained more precise estimates for the number of beeps than for the number of flashes.

Based on Bayesian probability theory, observers should therefore assign a stronger weight to the more precise auditory signal when integrating audiovisual signals into number estimates, resulting in the well-known sound-induced flash illusion [27,56,57]. Consistent with this conjecture, both SCZ and HC were more likely to perceive 2 flashes when a single flash was presented together with 2 sounds (i.e., fission illusion) and a single flash when 2 flashes were presented together with 1 sound (i.e., fusion illusion; see S3 Fig). We quantified these audiovisual interactions using the crossmodal bias (CMB) that ranges from pure visual (CMB = 1) to pure auditory (CMB = 0) influence (Fig 1D; n.b. the crossmodal bias can be computed only for numerically disparate flash-beep sequences). Consistent with the principles of precision-weighted integration [58], the HC and SCZ's flash reports were biased towards the number of auditory beeps (i.e., CMB < 1, $t_{39}$ = −11.864, $p < 0.001$, Cohen's d = −1.876, $BF_{10} > 100$), again with no significant differences between the groups ($t_{38}$ = 0.434, $p = 0.667$, d = 0.139, $BF_{10} = 0.336$). By contrast, we observed only a small but significant crossmodal bias for auditory reports towards the number of flashes in HC and SCZ (Fig 1D; CMB > 0, $t_{39}$ = 8.550, $p < 0.001$, d = 1.352, $BF_{10} > 100$)—again with no evidence for group differences ($t_{38}$ = 0.448, $p = 0.657$, d = 0.143, $BF_{10} = 0.337$). Thus, both HC and SCZ assigned a greater weight to the temporally more reliable auditory sense.

An additional 2 × 3 × 2 mixed-model ANOVA with factors group (SCZ versus HC), audiovisual numeric disparity between beeps and flashes (1, 2 versus 3), and task-relevance (auditory versus visual report) revealed that these crossmodal biases were significantly decreased at large numeric disparities, when signals most likely originated from different sources and should hence be segregated (i.e., task relevance × numeric disparity interaction, Table 2). At large numeric disparities, HC and SCZ were thus able to selectively report the number of flashes (or beeps) with minimal interference from task-irrelevant beeps (or flashes). This task relevance × numeric disparity interaction was qualitatively the key response profile predicted by BCI. It basically demonstrated that observers integrated signals at small conflict size, but segregated signals at large numeric conflicts when it was unlikely that auditory beep and visual flash sequences were generated by one common underlying source. Again, Bayes factors indicated substantial to strong evidence for comparable performance in SCZ and HC (i.e., $BF_{incl} < 1/3$ or even < 1/10, Table 2).

Overall, these GLM-based analyses of behavioral data (i.e., accuracy, crossmodal bias) suggested that both SCZ and HC combine audiovisual signals into number estimates qualitatively consistent with the principles of BCI. Both groups gave a stronger weight to the more reliable auditory signals leading to the well-known sound-induced flash illusions. Moreover, they arbitrated between sensory integration and segregation depending on the numeric disparity.

## Behavior—Bayesian modeling

To quantitatively assess whether HC and SCZ combine audiovisual signals according to shared computational principles, we compared 10 Bayesian models in a 5 × 2 factorial model space spanned by the modeling factors of "decision strategy" (5 levels) and "sensory variance" (2 levels) [28,30,31]. All models conformed to the basic architecture of the BCI model (Fig 3).

Along the factor "decision strategy," we manipulated how observers combined the estimates formed under fusion (i.e., common source) and segregation (i.e., separate sources) assumptions into a final perceptual estimate. While growing research has shown that HC combine audiovisual signals according to the decision strategy of model averaging [27,28,36], we hypothesized that SCZ may resort to suboptimal strategies or even simpler heuristics such as

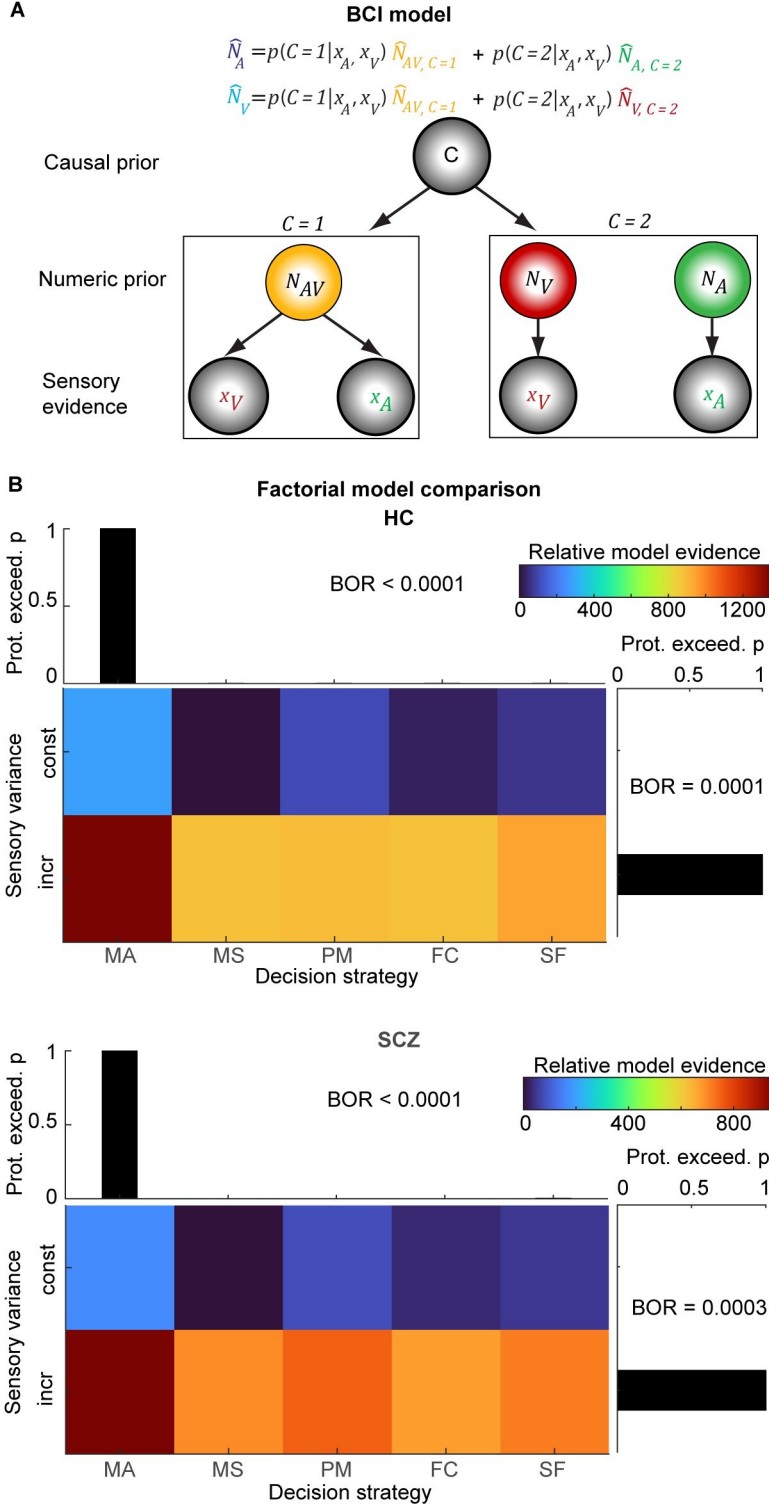

**Fig 3. The BCI model and factorial model comparisons in HC and SCZ. (A)** The BCI model assumes that audiovisual stimuli are generated depending on a causal prior ($p_{Common}$): In case of a common cause (C = 1), the "true" number of audiovisual stimuli ($N_{AV}$) is drawn from a common numeric prior distribution (with mean $\mu_P$) leading to noisy auditory ($x_A$) and visual ($x_V$) inputs. In case of independent causes (C = 2), the "true" auditory ($N_A$) and visual ($N_V$) numbers of stimuli are drawn independently from the numeric prior distribution. To estimate the number of auditory and visual stimuli given the causal uncertainty, the BCI model estimates the auditory or visual stimulus

number ($\hat{N}_A$ or $\hat{N}_V$, depending on the sensory modality that needs to be reported). In the model-averaging decision strategy, the BCI model combines the forced-fusion estimate of the auditory and visual stimuli ($\hat{N}_{AV,C=1}$) with the task-relevant unisensory visual ($\hat{N}_{V,C=2}$) or auditory estimates ($\hat{N}_{A,C=2}$), each weighted by the posterior probability of a common (C = 1) or independent (C = 2) causes, respectively (i.e., $p(C = 1|x_A, x_V)$ or $p(C = 2|x_A, x_V)$). **(B)** The factorial Bayesian model comparison (*n* = 40) of models with different decision strategies (model averaging, MA; model selection, MS; probability matching, PM; fixed criterion, FC; stochastic fusion, SF) with constant or increasing sensory auditory and visual variances, separately for HC and SCZ. The images show the relative model evidence for each model (i.e., participant-specific Bayesian information criterion of a model relative to the worst model summed over all participants). A larger model evidence indicates that a model provides a better explanation of our data. The bar plots show the protected exceedance probability (i.e., the probability that a given model is more likely than any other model, beyond differences due to chance) for each model factor. The BOR estimates the probability that factor frequencies purely arose from chance. Source data is provided in S3 Data. BCI, Bayesian causal inference; BOR, Bayesian omnibus risk; HC, healthy control; SCZ, schizophrenia.

applying a fixed threshold on audiovisual numeric disparity [31]. In other words, rather than arbitrating between sensory integration and segregation according to the posterior probability of a common cause in a Bayesian fashion, SCZ may simply do so based on audiovisual spatial disparity.

In total, we compared the following 5 decisional strategies: (i) Model averaging combines the segregation and fusion estimates weighted by the posterior probabilities of each causal structure. For example, it gives a stronger weight to the fusion estimate when it is likely that the flash and beep sequences are generated by one common cause. (ii) Model selection selects the perceptual estimate from the causal structure with the highest posterior probability. So rather than averaging fusion and segregation estimates, it selectively reports either the fusion or segregation estimates depending on the posterior probabilities of common and independent causes. (iii) Probability matching selects either the fusion or segregation estimates in proportion to their posterior probability. (iv) The fixed-criterion threshold model incorporates the simple heuristic of selecting the segregation estimate when the audiovisual numeric disparity estimate exceeded a fixed threshold. This differs from Bayesian model selection strategies in that observers' do not apply the threshold to the posterior probability of a common cause, but directly on the numeric disparity estimate—thereby ignoring observers' uncertainty about this estimate. (v) Finally, the stochastic fusion model employs the suboptimal strategy of selecting either the fusion or segregation estimates with a fixed probability that is estimated from observers' responses (see Materials and methods for details and [31]).

Additionally, we assessed along the modeling factor "sensory variance" whether auditory and visual noise variances were constant or changed with the number of beeps/flashes as predicted by the scalar-variability of numerosity estimation [52,53] and initial inspection of our data (see Figs 2 and S1).

Comparing all 10 models in this 2 × 5 factorial model space (Fig 3 and S2 Table) revealed that the model-averaging model with increasing sensory variances outperformed all other 9 models in both HC and SCZ observers. Furthermore, a between-group Bayesian model-comparison provided strong evidence that HC and SCZ individuals relied similarly on the 5 decision strategies ($BF_{10}$ = 0.027). In particular, model-averaging turned out as the "winning" strategy equally often (Fig 3B). These Bayesian model-comparison results further supported the notion that SCZ, like HC, performed multisensory perceptual and causal inference according to the same computational and decision strategies. S4 Fig also shows the BICs as approximations to the model evidence for each of the 5 models that allowed for scalar variability for each patient ranked according to their PANSS score. The figure shows that the model averaging was the dominant strategy in almost all of the patients irrespective of their psychosis severity.

Having established that SCZ and HC combined audiovisual signals in line with Bayesian causal inference and read out the final estimate according to model averaging, we examined

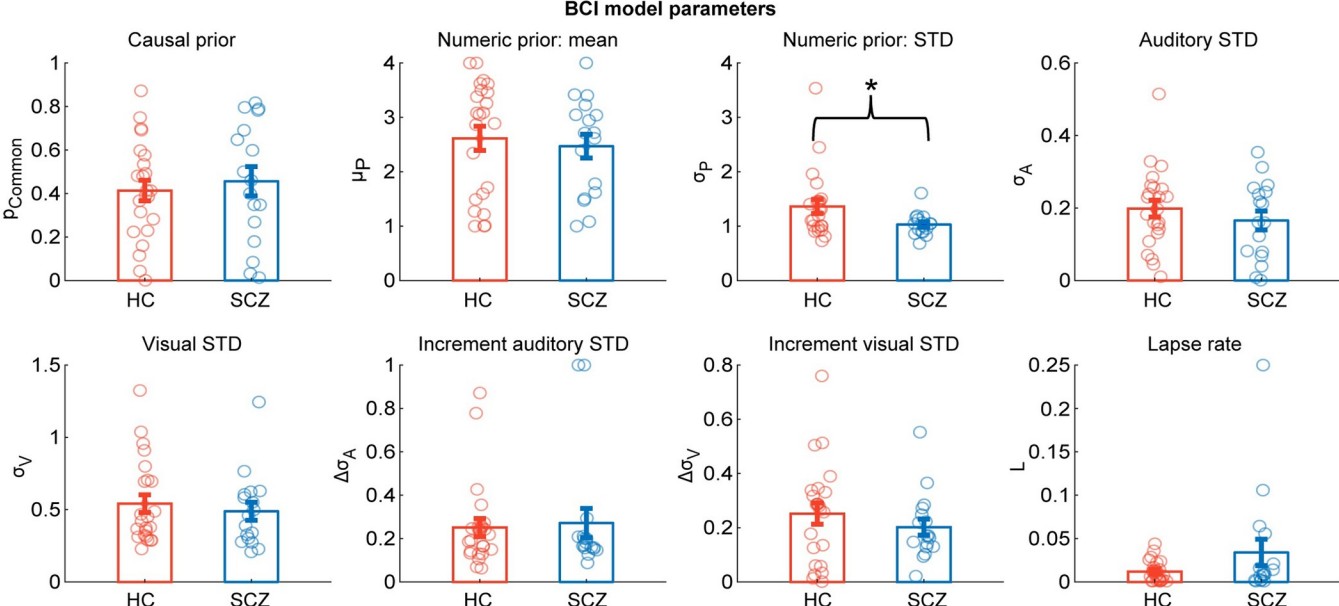

**Fig 4. The parameters of the BCI model (across participants mean ± SEM; *n* = 40) separately plotted for HC and SCZ patients.** The BCI model's decision strategy applies model averaging with increasing sensory variance. Significant different are indicated by * = $p < 0.05$. Source data is provided in S4 Data. BCI, Bayesian causal inference; HC, healthy control; SCZ, schizophrenia.

whether SCZ may overweight their causal prior about the signals' causal structure (i.e., $p_{common}$). Contrary to this conjecture, two-sample randomization tests on $p_{common}$ did not reveal any significant difference between the 2 groups (Fig 4 and Table 3). In the next step, we asked whether SCZ relied more strongly on their priors about the flash/beep number relative to the sensory inputs as may be hypothesized based on a growing number of studies in unisensory perception [10,11,13,15]. An over-reliance on prior information would be reflected in a greater precision (i.e., smaller variance; $\sigma_P$) of numeric priors (i.e., $\mu_P$) in SCZ relative to HC. As shown in Fig 4 and Table 3, two-sample randomization tests indeed revealed a significantly smaller variance for the numeric prior in SCZ compared to HC. This more precise numeric prior accounts for the fact that the numeric reports are centered more around the numeric prior mean in SCZ compared to HC (Table 3; cf. Fig 2). It is consistent with previous research [10,11,13,15] and consistent with the general notion that SZC rely more on prior knowledge than new incoming audiovisual evidence. However, the effect was small and did not survive correction for multiple

**Table 3. Comparison of the BCI model's parameters between HC and SCZ participants.**

|  | $p_{common}$ | $\mu_P$ | $\sigma_P$ | $\sigma_A$ | $\sigma_V$ | $\Delta\sigma_A$ | $\Delta\sigma_V$ | L |
|---|---|---|---|---|---|---|---|---|
| $t_{38}$ | −0.534 | 0.454 | 2.109 | 0.938 | 0.599 | −0.275 | 0.958 | −1.658 |
| p | 0.593 | 0.653 | 0.026 | 0.366 | 0.553 | 0.802 | 0.331 | 0.062 |
| $p_{corr}$ | 0.731 | 0.731 | 0.210 | 0.717 | 0.731 | 0.811 | 0.717 | 0.282 |
| Cohen's d | −0.171 | 0.145 | 0.675 | 0.300 | 0.191 | −0.088 | 0.306 | −0.530 |
| $BF_{10}$ | 0.349 | 0.338 | 1.725 | 0.441 | 0.359 | 0.321 | 0.448 | 0.909 |

Note: The BCI model's decision strategy applies model averaging with increasing sensory variance; *p*-values derived from a two-sample two-sided randomization test (*n* = 5,000 randomizations). $p_{corr}$ is multiple-comparison corrected across the 8 parameters using the Benjamini–Hochberg correction for false-discovery rate. $p_{common}$, causal prior; $\mu_P$, mean of the numeric prior; $\sigma_P$, standard deviation of the numeric prior; $\sigma_A$, standard deviation of the auditory likelihood; $\sigma_V$, standard deviation of the visual likelihood; $\Delta\sigma$, increase of standard deviation per auditory or visual signal number; L, lapse parameter.

comparisons across the model parameters or when including the 6 additional patients with schizoaffective disorder (cf. S1 Text and S7 Table).

Further, we observed a significant positive correlation between SCZ's PANSS positive symptoms with their visual variance (r = 0.625, $t_{16}$ = 3.101, *p* = 0.003, $p_{corr}$ = 0.024, $BF_{10}$ = 6.518; randomization test of the correlation). This effect remained significant even after controlling for PANSS negative symptoms and general psychopathology (S3 Table). Further, we correlated the BCI model's parameters with self-reported hallucinatory experiences (i.e., LSHS-R) and paranoid thinking (i.e., PCL) as sensitive measures of psychosis severity that may also capture subclinical psychotic symptoms. LSHS-R and PCL were associated with greater variability in our sample compared to the PANSS positive symptoms (S5 Fig). Only the auditory variance correlated negatively with hallucinatory experiences at marginal significance (without multiple comparison correction, r = −0.498, $t_{16}$ = −2.223, *p* = 0.047, $p_{corr}$ = 0.141, $BF_{10}$ = 1.439; S3 Table). In addition, we observed a positive correlation between PANSS negative symptoms and lapse rates in SCZ (r = 0.483, $t_{16}$ = 2.134, *p* = 0.029, $p_{corr}$ = 0.232, $BF_{10}$ = 1.248). No other significant group differences were revealed when comparing the BCI model's parameters between the 2 groups (Fig 4 and Table 3).

These quantitative Bayesian modeling results corroborated our initial GLM-based conclusions that both HC and SCZ combined audiovisual signals consistent with Bayesian causal inference. The numeric prior was slightly more precise in SCZ compared to HC (according to classical statistics), which was in line with previous results in unisensory perception [10,11,13,15]. Moreover, PANSS positive symptoms correlated positively with visual variance and the LSHS-R correlated negatively with auditory variance. Collectively, these results suggested that patients with psychotic symptoms may have relied more strongly on prior information and possibly auditory information relative to visual inputs for numeric inferences. The positive correlation between PANSS negative symptoms and lapse parameter indicated that at least some SCZ patients may have found it harder to stay attentive throughout the entire experiment.

## Behavior—Adjustment of priors

To further explore whether SCZ over-relied on prior information, we exploited the fact that observers dynamically adapt their priors in response to previous stimuli. Some previous studies have suggested that this dynamic updating of priors is altered in psychosis ([10], but see [18]). Thus, we first examined whether and how SCZ and HC increased their binding tendency ("model-free") or causal prior (i.e., $p_{common}$ from Bayesian modeling analysis) after exposure to congruent or incongruent flash-beep sequences. As expected based on previous findings [27,59–61], both SCZ and HC similarly decreased their binding tendency after a trial with greater numeric disparity. As shown in S6A Fig, the differences between the CMB for visual and auditory report increased with the greater numerical disparity of the previous trial. In other words, observers were more capable to selectively report the numeric estimate of the task-relevant sensory modality for large numeric disparities on the previous trials. This was consistent with the idea that after a large numeric disparity trial, observers decreased their prior binding tendency which in turn attenuated audiovisual interactions and crossmodal bias on subsequent trials. Likewise, our Bayesian modeling analysis indicated that observers' causal prior increased after a congruent trial and decreased after a trial with large numeric disparity (i.e., a main effect of previous disparity; Fig 5A and Table 4). Importantly, we did not observe any significant differences between groups (i.e., no significant disparity × group interaction).

Next, we investigated the influence of the number of signals on the previous trial on observers' numeric estimates. As expected, observers' numeric estimates were biased towards the

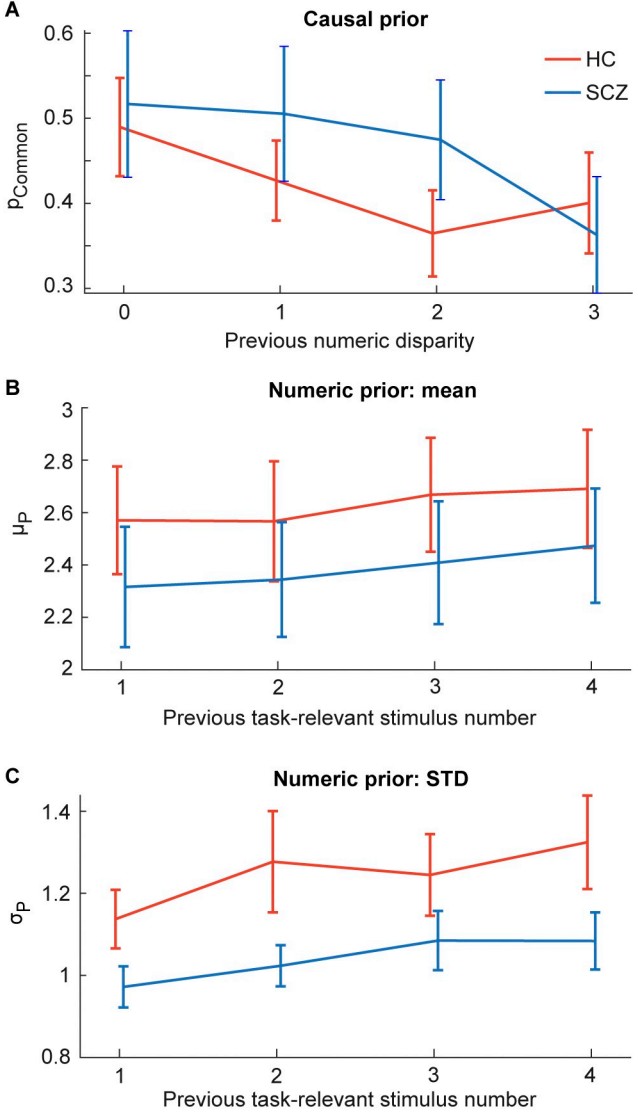

**Fig 5. Adjustment of the BCI model's causal and numeric priors (mean ± between-participants' SEM; *n* = 40) in the current trial to the previous trial's numeric disparity or task-relevant stimulus number in HC and SCZ. (A)** The current causal prior ($p_{Common}$) as a function of previous audiovisual numeric disparity (i.e., $|n_A - n_V|$). **(B)** The current numeric prior's mean ($\mu_P$) as a function of the previous task-relevant stimulus number (i.e., $n_A$ for auditory and $n_V$ for visual report). **(C)** The current numeric prior's STD ($\sigma_P$) as a function of the previous task-relevant stimulus number. Source data is provided in S5 Data. BCI, Bayesian causal inference; HC, healthy control; SCZ, schizophrenia.

number of signals on the previous trial (see S6B Fig). Again, this was also reflected in our Bayesian modeling analysis assessing the updating of observers' numeric prior (i.e., $\mu_P$) and its variance (i.e., $\sigma_P$). As expected for Bayesian learners, both SCZ and HC increased their numeric prior's mean and variance after exposure to a high number of signals in the task-relevant modality (e.g., beeps for auditory report; Fig 5B and 5C; significant main effects in Table 4). But again, no significant differences were observed between groups (i.e., no significant interaction effects with group) suggesting that HC and SCZ also adjust their perceptual priors at shorter timescales similarly.

In summary, our analyses of the behavioral data based on the general linear model (GLM) and formal Bayesian modeling suggest that the medicated SCZ dynamically adapted and

**Table 4. Main and interaction effects of previous audiovisual disparity (Disp) or audiovisual task-relevant stimulus number (Stim#) and group on the BCI model's current causal prior ($p_{common}$), the numeric prior's mean ($\mu_P$) and STD ($\sigma_P$) from classical and Bayesian mixed-model ANOVAs.**

| Parameter | | F | df1, df2 | p | part. $\eta^2$ | BF$_{incl}$ |
|---|---|---|---|---|---|---|
| $p_{common}$ | Disp | 5.949 | 3, 114 | <0.001 | 0.135 | 17.903 |
| | Group | 0.291 | 1, 38 | 0.593 | 0.008 | 0.571 |
| | Disp×Group | 2.195 | 3, 114 | 0.103 | 0.055 | 0.852 |
| $\mu_P$ | Stim# | 4.820 | 2.6, 97.3 | 0.006 | 0.113 | 7.382 |
| | Group | 0.568 | 1, 38 | 0.456 | 0.015 | 0.574 |
| | Stim#×Group | 0.127 | 2.6, 97.3 | 0.923 | 0.003 | 0.130 |
| $\sigma_P$ | Stim# | 6.169 | 3, 114 | <0.001 | 0.140 | 53.296 |
| | Group | 2.706 | 1, 38 | 0.108 | 0.066 | 0.786 |
| | Stim#×Group | 0.924 | 3, 114 | 0.432 | 0.024 | 0.361 |

Note: Disp = $|n_A - n_V|$; Stim# = $n_A$ or $n_V$ depending on their task relevance; Group: HC vs. SCZ; effects of the classical mixed-model ANOVAs are Greenhouse–Geisser corrected if sphericity is violated.

combined priors about the signals' causal structure and the number of signals with new audiovisual inputs comparable to the healthy controls and in line with Bayesian principles.

## EEG—Decoding of numeric Bayesian causal inference estimates

To investigate whether SCZ and HC achieve Bayesian causal inference via shared or different neural processes, we combined BCI models with EEG decoding. In particular, we temporally resolved how the brain encodes the auditory ($\hat{N}_{A,C=2}$) and visual ($\hat{N}_{V,C=2}$) segregation, the fusion ($\hat{N}_{AV,C=1}$) and the final BCI perceptual estimates (i.e., $\hat{N}_A$ or $\hat{N}_V$) that combine the forced-fusion estimate with the task-relevant unisensory segregation estimates, weighted by the posterior causal probability of common or separate causes [27]. To track the evolution of these different estimates across time, we trained a linear support-vector regression model (SVR decoder) to decode each of the 4 BCI estimates from EEG patterns of 60 ms time windows. SVR decoders were trained on EEG patterns individually in each participant. We computed the decoding accuracy as the Pearson correlation between the decoded and "true" perceptual estimates to quantify how strongly a perceptual estimate was encoded in EEG patterns for each participant. Decoding performance at chance level was defined as a correlation of zero (i.e., r = 0). At the group level, we then assessed whether the decoded perceptual estimates (as indicated by the correlation coefficients) evolved with different time courses in HC and SCZ.

In both HC and SCZ, the decoders predicted the BCI estimates from EEG patterns significantly better than chance over most periods of the post-stimulus time window (i.e., decoding accuracy r ≈ 0.3; Fig 6A). Yet, the different perceptual estimates evolved with different time courses commonly in both groups. Initially, the EEG activity encoded mainly the visual segregation estimates starting at about approximately 60 to 100 ms (i.e., significant clusters in one-sided cluster-based corrected randomization t test; see S4 Table). Slightly later, the auditory segregation and fusion estimates peaked with the fusion estimates showing a slower decline. The final BCI estimate, which accounts for the signals' causal structure, rose more slowly and showed a more sustained time course. Moreover, in HC only did the decoding accuracy of the final BCI estimate exceed those of the other estimates at 600 ms. The temporal profiles of the decoding accuracies were largely comparable between HC and SCZ. Bayes factors provided mainly weak evidence for no difference between the groups (Fig 6B), except for the visual segregation-estimate that was associated with a significantly lower and more protracted decoding

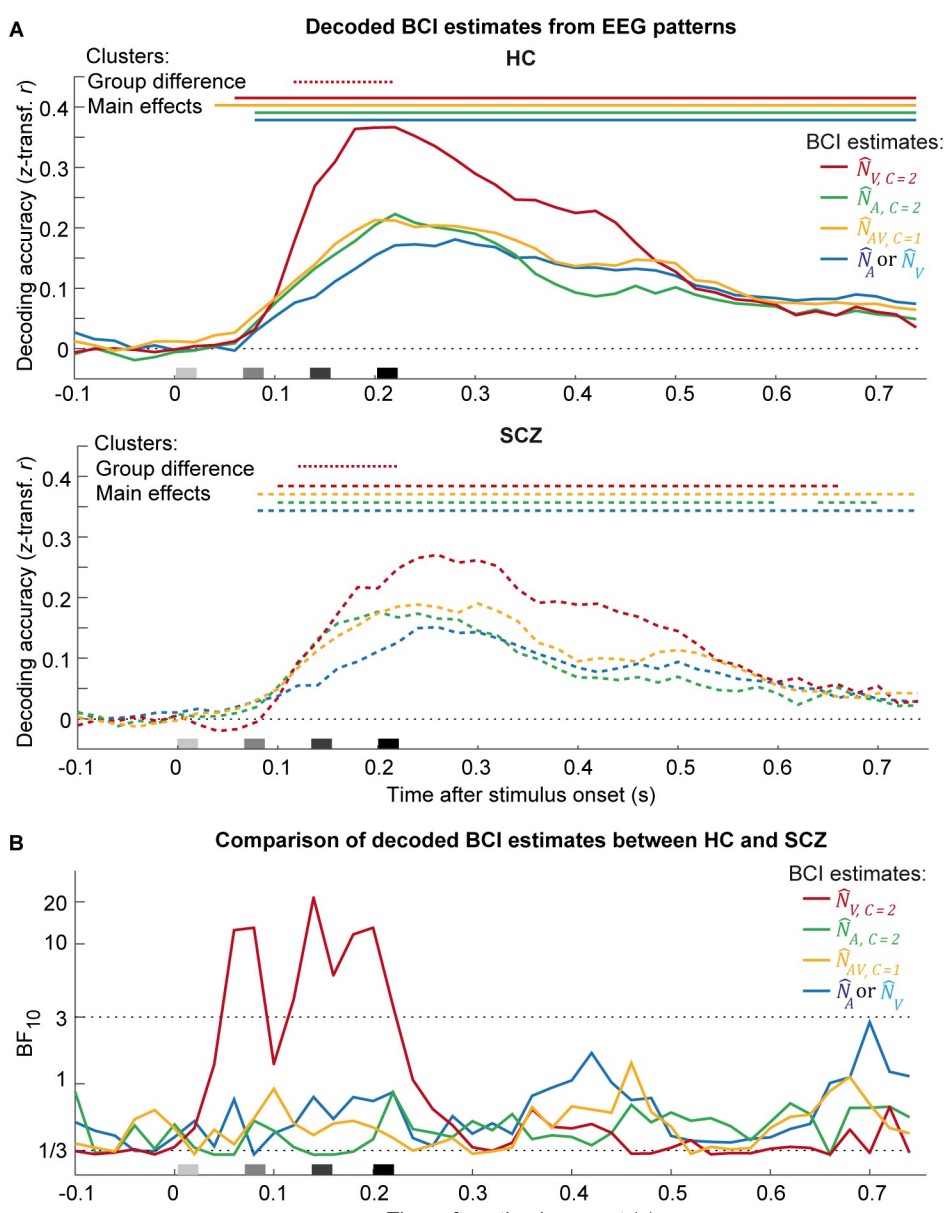

**Fig 6. Decoding the BCI model's numeric estimates from EEG patterns using SVR in HC versus SCZ ($n$ = 40). (A)**
Decoding accuracy (Fisher's $z$-transformed correlation; across-participants mean) of the SVR decoders as a function of time and group (HC vs. SCZ). Decoding accuracy was computed as correlation coefficient between the given BCI model's internal estimates and BCI estimates that were decoded from EEG activity patterns using SVR models trained separately for each numeric estimate. The BCI model's internal numeric estimates comprise of: (i) the unisensory visual ($\hat{N}_{V,C=2}$), (ii) the unisensory auditory ($\hat{N}_{A,C=2}$) estimates under the assumption of independent causes (C = 2), (iii) the forced-fusion estimate ($\hat{N}_{AV,C=1}$) under the assumption of a common cause (C = 1), and (iv) the final BCI estimate ($\hat{N}_A$ or $\hat{N}_V$ depending on the sensory modality that is task-relevant) that averages the task-relevant unisensory and the precision-weighted estimate by the posterior probability estimate of each causal structure. Color-coded horizontal solid lines (HC) or dashed lines (SCZ) indicate clusters of significant decoding accuracy ($p < 0.05$; one-sided one-sample cluster-based corrected randomization $t$ test). Color-coded horizontal dotted lines indicate clusters of significant differences of decoding accuracy between both groups ($p < 0.05$; two-sided two-sample cluster-based corrected randomization $t$ test). Stimulus onsets are shown along the x-axis. **(B)** Bayes factors for the comparison between the decoding accuracies of HC and SCZ for each of the BCI estimate (i.e., $BF_{10} > 3$ substantial evidence for or $BF_{10} < 1/3$ against group differences). Source data is provided in S6 Data. BCI, Bayesian causal inference; EEG, electroencephalography; HC, healthy control; SCZ, schizophrenia; SVR, support-vector regression.

accuracy in SCZ than HC from 120 to 220 ms (Fig 6A and S4 Table; all further clusters $p > 0.05$). Because we did not track observers' eye movements, these lower decoding accuracies in SCZ may potentially be explained by reduced fixation stability in SCZ [62]. Potentially, it may also result from increases in visual uncertainty in SCZ patients that were reported in previous studies on the computational mechanisms of causal inference. These previous studies manipulated the asynchrony of audiovisual stimuli in simultaneity judgment tasks [42,63] and may therefore have been more sensitive to detect differences in sensory uncertainty than the current study that categorically manipulated the number of beeps and flashes. In summary, combining Bayesian modeling and EEG decoding largely confirmed that SCZ and HC performed audiovisual number estimation according to similar neurocomputational mechanisms. However, SCZ encoded the visual signal with less precision resulting in a lower decoding accuracy of the visual segregation estimate. These results suggested that the visual uncertainty may be increased in SCZ participants as previously reported [42]. By characterizing sensory uncertainty across time at millisecond resolution, EEG may be able to reveal sensory differences that may not be reflected in observers' behavioral responses.

## EEG–Multisensory interactions in ERPs

Following previous work, we also analyzed and compared the event-related potentials (ERPs) to unisensory and multisensory stimuli between HC and SCZ [46,64–68]. ERPs showed the typical components in response to flashes and beeps (Fig 7), i.e., P1 (approximately 50 ms), N1 (100 ms), P2 (200 ms), N2 (280 ms), and P3 (>300 ms) [69]. As previously reported [70], the unisensory visual P3 component was significantly smaller in SCZ than HC (cluster 560 to 705 ms, $p = 0.045$). To characterize the neural processes of audiovisual integration, we tested for audiovisual interactions (i.e., $AV_{congr}$ versus $(A + V)$) over occipital electrodes. In line with

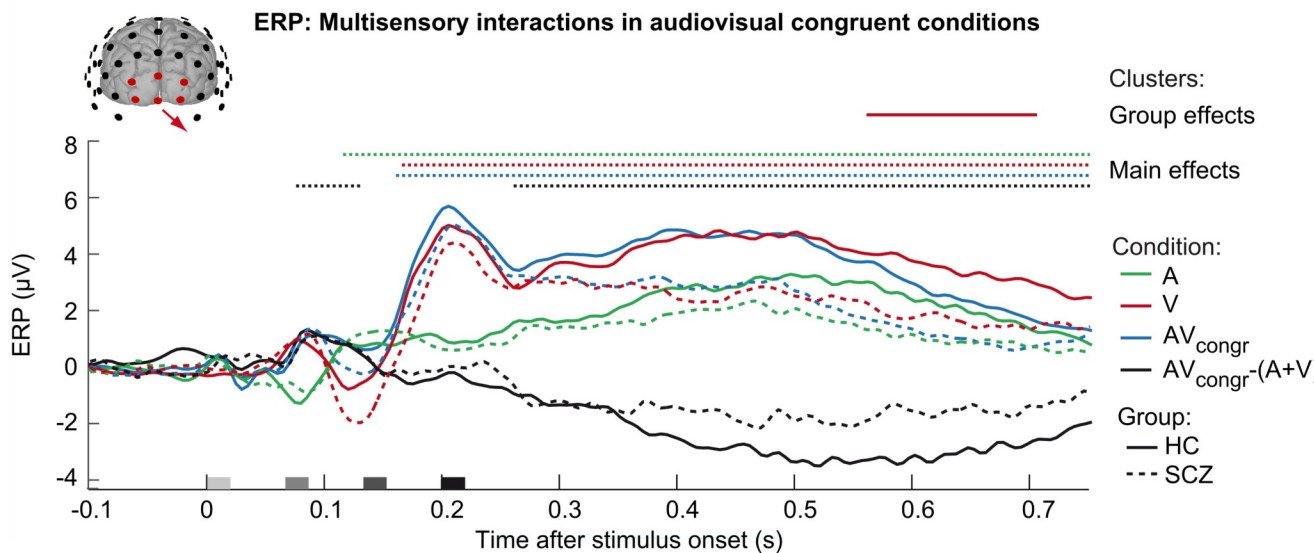

**Fig 7. Occipital ERPs in response to unisensory and audiovisual stimuli in HC and SCZ.** ERPs (across-participants mean grand averages; $n = 40$) of HC and SCZ participants elicited by unisensory auditory stimuli (A), unisensory visual stimuli (V), audiovisual congruent conditions ($AV_{congr}$) and the difference of these ERPs (i.e., $AV_{congr}$−(A+V)) indicating multisensory interactions. The ERPs are averaged across occipital electrodes. Color-coded horizontal dotted lines indicate significant clusters ($p < 0.05$) of ERPs against baseline (i.e., across HC and SCZ, main effect of condition) in one-sample two-sided cluster-based corrected randomization t tests. The horizontal solid line indicates a significant cluster of ERP differences between HC and SCZ in two-sample two-sided cluster-based corrected randomization t tests. The x-axis shows the stimulus onsets. Source data is provided in S7 Data. ERP, event-related potential; HC, healthy control; SCZ, schizophrenia.

previous studies [69,71], we observed early audiovisual interactions from 75 to 130 ms (i.e., measured from the onset of the first flash-beep slot; $p = 0.049$; two-sided one-sample cluster-based corrected randomization t test) and later negative audiovisual interactions from 260 to 750 ms after stimulus onset ($p < 0.001$). Crucially, the ERP interactions did not differ between HC and SCZ ($p > 0.05$; two-sided two-sample cluster-based corrected randomization test).

To examine whether SCZ patients and HC may differ more subtly in their multivariate EEG response, we trained a support-vector classifier (SVC) to classify participants [72] as SCZ or HC based on their EEG activation patterns to auditory, visual, and audiovisual stimuli across poststimulus time windows (see S1 Text and S7 Fig). The ERP topographies evolved largely similar in HC and SCZ, so that the EEG-based decoder did not classify participants as HC or SCZ better than chance in one-sided cluster-based corrected randomization tests when corrected across multiple comparisons across time. Overall, our ERP and multivariate analyses thus corroborated that the neural mechanisms underlying audiovisual integration were largely comparable in HC and SCZ patients. However, in line with our EEG results from the BCI analysis, we observed a deficit in visual processing in SCZ as indicated by smaller visual P3 responses.

## Discussion

This study combined psychophysics, EEG, and Bayesian modeling to investigate whether and how schizophrenia impacts the computational and neural mechanisms of multisensory causal and perceptual inference. A growing number of studies suggests that schizophrenia may alter the brain's ability to integrate audiovisual signals into coherent percepts (for a review, see [44]). Many of these studies have used qualitative approaches such as evaluating the frequency of perceptual illusions in schizophrenia compared to healthy controls. Yet, the complex dependencies of multisensory illusion rates on sensory precisions and perceptual as well as causal priors may have contributed to the inconsistent findings reported so far [41,43,46]. In this study, we have therefore employed a more sophisticated factorial design with formal Bayesian modeling to disentangle these different computational components.

Our GLM-based and Bayesian modeling analyses offered convergent evidence that SCZ as their HC counterparts performed multisensory perceptual and causal inference in line with normative principles of Bayesian causal inference. When numeric disparities were small and signals were likely to originate from common sources, individuals with SCZ and HC integrated auditory and visual signals weighted by their relative precisions resulting in crossmodal biases. At large numeric disparities, these crossmodal biases and interactions were reduced. Through formal Bayesian model comparison, we further examined whether SCZ computed Bayesian estimates or resort to simpler heuristics such as segregating signals above a numeric disparity threshold. Our quantitative Bayesian modeling analyses confirmed that SCZ, like their healthy counterparts, combined signals consistent with BCI models and read out final numeric estimates based on the decisional strategy of model averaging.

Next, we examined whether SCZ may overweight prior information about flash/beep number in relation to sensory evidence at long and/or short timescales as previously observed in unisensory perception [10,11,13,15]. In support of an overweighting of priors in psychosis, our Bayesian modeling analysis revealed a significantly more precise numeric prior for SCZ compared to HC. Further, the visual variance correlated positively with the positive symptoms on the PANSS scale. These findings are consistent with previous studies showing overreliance of patients with psychosis on prior knowledge relative to sensory evidence [10–15]. However, the effect on prior variance was very small. It was no longer significant after correcting for multiple comparisons or including schizoaffective patients (cf. S1 Text and S7 Table). Likewise, the

influence of the previous trial's flash/beep number and numeric disparity on current perceptual choices was comparable in SCZ and HC.

Furthermore, we cannot exclude the possibility that SCZ showed a bias towards an intermediate signal number, which could be misinterpreted as overreliance on their numeric prior (see Figs 4 and 5C), because they were less vigilant or motivated and therefore failed to count some signals. Indeed, in line with this notion, the lapse parameter correlated positively with SCZs' negative symptoms (S3 Table) and negatively with SCZs' memory performance in the VLMT as well as their ability to inhibit inappropriate responses in the Stroop test (S5 Table). Overall, our analyses thus suggested that the medicated SCZ group of our study weighted and dynamically adapted numeric priors in a largely comparable fashion as their healthy counterparts with a small trend towards an overreliance on their numeric prior relative to bottom-up audiovisual evidence. The absence of substantial prior overweighting may be explained by our specific sample of medicated patients with chronic rather than acute SCZ and low expression of psychotic symptoms. Thus, prior overweighting may be a state marker associated with psychotic symptoms rather than a trait marker for schizophrenia [12].

Moreover, to our knowledge, this is the first study assessing prior weighting in a multisensory context. The availability of pieces of evidence furnished simultaneously by 2 sensory modalities may decrease the overall reliance on prior information and may thus decrease our paradigm's sensitivity to detect small modulations in prior variance and weighting. Further, it is conceivable that distinct mechanisms across the cortical hierarchy support integration of prior and sensory evidence depending on whether information is provided only in one or in several sensory modalities. Given the prevalence of multisensory hallucinations in patients with psychosis [21], future work is required to assess perceptual inference in multisensory situations. For example, one may even develop elegant multisensory conditioning paradigms to provoke conditioned audiovisual hallucinations [11].

Crucially, our multisensory paradigm enabled us to assess not only the weighting of prior knowledge regarding environmental properties such as flash/beep number, but also to the world's causal structure, i.e., whether signals come from common or independent sources as incorporated in the causal prior. A high causal prior or binding tendency increases crossmodal biases and interactions, while a low causal prior enhances an individual's ability to selectively report the signal number in the task-relevant sensory modality while ignoring the incongruent number of signals in the irrelevant sensory modality. Changes in the causal prior may thus be closely associated with conflict monitoring, cognitive control, and selective attention mechanisms [51,73–75]. Yet, despite growing evidence for impaired selective attention and cognitive control mechanisms in SCZ [76–78], our Bayesian modeling analyses did not reveal any significant differences in the causal prior between SCZ and HC.

In summary, our GLM-based and Bayesian modeling analyses showed that SCZ and HC combined perceptual and causal priors with sensory evidence in a manner consistent with the principles of normative Bayesian causal inference. Furthermore, Bayesian statistics provided consistent evidence that the causal and perceptual priors' weighting and updating was largely maintained in the SCZ group. These findings suggest that both groups dynamically adapt to the statistical structure of multisensory stimuli across short time scales (cf. serial dependencies in perception [79]). The absence of notable computational abnormalities in SCZ could be explained by the fact that our patient group was post-acute and medicated and scored low on the PANSS positive symptom scale. Most importantly, our cohort of SCZ patients also showed near-normal performance on the TMT-B and Stroop tests (cf. Table 1). Both tests measure executive and attentional functions which are particularly relevant for optimal performance in our inter-sensory selective-attention paradigm. Our findings add further evidence to the fact that fundamental mechanisms of perceptual inference are preserved at least in subgroups of medicated SCZ individuals [16–19]. Future

research is required to explore whether schizophrenia patients with attentional and executive deficits and in more severe psychosis states may exhibit abnormal causal priors, and thus, how they arbitrate between sensory integration and segregation.

Using EEG, we investigated whether SCZ and HC support these computations via similar or different neural mechanisms. For example, in SCZ, the computations may involve compensatory neural systems or exhibit a slower dynamic. To explore these questions, we decoded the auditory and visual segregation, the fusion and the final BCI model's estimates from scalp EEG data. In HC, the decoding accuracy of the visual segregation-estimate peaked earlier and higher compared to the other estimates. By contrast, in the SCZ group, decoding of the visual estimate showed a more protracted time course initially overlapping with the auditory segregation and fusion estimates. Statistical tests confirmed a significant difference in decoding accuracy between HC and SCZ from approximately 100 to 200 ms. However, apart from this difference in decoding of the unisensory visual estimate and similarly in the visual P3 ERP component, the decoding profiles appeared largely similar across the 2 groups.

To ensure that we did not miss any differences between SCZ and HC that were previously reported in the literature [46,64–68], we also performed standard ERP analyses to test for audiovisual interactions. However, these analyses revealed only an attenuated visual P3 component in SCZ, but no significant differences in audiovisual interactions between the 2 groups. Similarly, we were unable to predict group membership (i.e., HC versus SCZ) from multivariate audiovisual EEG responses significantly better than chance.

To conclude, our behavioral, computational, and neuroimaging results consistently demonstrate that audiovisual perception in a sound-induced flash illusion paradigm is based on comparable computational and neural mechanisms in SCZ and HC. Both SCZ and HC combined audiovisual signals into number estimates in line with the computational principles of Bayesian causal inference. However, our small sample size may have prevented us from detecting more subtle potential alterations in SCZ (cf. see limitations below). Our computational modeling approach moves significantly beyond previous studies assessing audiovisual illusion rates by characterizing patients according to their perceptual inference, decisional strategy, and specific parameters in the winning model. Additional time-resolved EEG decoding revealed that both HC and SCZ performed Bayesian causal inference by dynamically encoding the visual and auditory segregation- and fusion-estimates followed by later estimates that flexibly integrate auditory and visual information according to their causal structure. Collectively, our results thus showed that at least in our limited sample of post-acute medicated SCZ patients, the computations and neural mechanisms of hierarchical Bayesian causal inference in audiovisual numeric perception were largely preserved.

However, several factors limit the scope of these conclusions: First, the sample size was small and only adequate to reveal large effect sizes. Our study was able to detect significant larger differences between SCZ and HC in the numeric prior (e.g., Cohen's d = 0.675) or larger correlations between the BCI model parameters and psychopathology such as between visual variance and participant's psychotic symptoms (e.g., r = 0.625). Yet, the small sample size may have precluded the detection of more subtle effects. Our choice of a small sample size was guided by previous research into behavioral multisensory deficits in SCZ that reported large effect sizes for differences between SCZ and controls in small sample sizes between 12 and 30 patients [40,80,81]. However, these previous small-sample studies may have led to an unrealistic overestimation of the effect size due to the "winner's curse" phenomenon, i.e., the tendency for published studies to report effect sizes that exceed the true effect size [82]. In fact, effect sizes of interindividual differences can generally be expected to be much smaller [83]. However, increasing the patient sample size by 6 patients with schizoaffective disorder yielded highly similar results for the CMB, BCI modeling and decoding of BCI estimates and even

turned the overreliance on the numeric prior nonsignificant (see S1 Text and S8–S11 Figs and S6–S8 Tables). Moreover, Bayes factors frequently supported the absence of group differences. Nevertheless, studies with higher power are needed to corroborate that our study did not miss out on subtle alterations in multisensory perceptual inference in SCZ. Second, as a result of our recruitment strategy, our sample included mainly chronic patients with high doses of antipsychotics, relatively intact cognitive functions and low psychosis severity (cf. Table 1 and S5 Fig). By contrast, previous studies reported that overweighting of perceptual priors arises specifically in individuals with acute hallucinations [14], independently from a diagnosis of schizophrenia [11]. Thus, prior overweighting could rather be a state-marker for psychotic episodes than a trait-marker for schizophrenia [12]. Our conclusions of relatively intact multisensory perceptual and causal inference may not hold for individuals with prodromal or acute psychotic states of schizophrenia or patients with persistent post-acute hallucinations. Thus, future studies are needed to assess multisensory perceptual inference in a larger sample of SCZ participants that vary substantially in their psychotic symptoms. It has even been suggested that psychotic symptoms arise on a symptomatic continuum across healthy individuals and patients [84], so that altered weighting of priors in multisensory causal and perceptual inference may be found in nonclinical samples with subclinical psychotic symptoms. In line with this conjecture, a recent study in young nonclinical participants reported that individuals with stronger prodromal psychotic symptoms showed reduced causal priors [50]. Third, our small clinical SCZ sample was still heterogeneous with respect to the duration of illness, comorbidities, the dose and type of antipsychotic medication, and the strength of acute positive and negative symptoms as well as cognitive impairments. This substantial heterogeneity might have obfuscated small alterations in multisensory perceptual and causal inference in SCZ. Fourth, our experimental SIFI paradigm manipulated the number of flashes and beeps to investigate causal inference via numeric disparities. Previous studies have shown a widening of the temporal binding window in SCZ ([42]; see [85] for a review), which may be further investigated in SIFI paradigms with variable audiovisual asynchronies. Indeed, one previous study [42] has varied the audiovisual asynchrony of audiovisual stimuli to assess Bayesian causal inference in SCZ, individuals with autism spectrum disorder (ASD) and controls. Interestingly, the widening of the temporal binding window which was common to both ASD and SCZ could be attributed to different computational parameters. While it resulted from a stronger causal prior or binding tendency in ASD, it arose from larger sensory uncertainty in SCZ. Manipulating audiovisual asynchrony enables more precise estimation of observers' sensory uncertainty than categorically manipulating the number of flashes and beeps as in our paradigm. Thus, it is possible that subtle increases in sensory uncertainty in SCZ may have gone undetected in our paradigm.

Future research is needed to determine whether deviations from normative Bayesian principles in multisensory perception may occur in larger samples of unmedicated psychotic patients in prodromal or acute stages or patient groups with more pronounced impairments on their attentional and executive functions. Critically, our study focused on simple artificial flash-beep stimuli. This makes it an important future research direction to investigate multisensory perceptual and causal inference in psychosis in more naturalistic situations such as face-to-face communication.

## Materials and methods

### Ethics statement

The study was performed in line with the principles of the Declaration of Helsinki, and it was approved by the human research review committee of the Medical Faculty of the University of Tuebingen and at the University Hospital Tuebingen (approval number 728/2014BO2).

## Participants

After giving written informed consent, 24 healthy volunteers and 26 post-acute in- and out-patients participated in the EEG study. This small sample size was approximately in line with power calculations that assumed large effect sizes (Cohen's d = 1.00) consistent with previous studies that reported large effect sizes for differences between SCZ and controls in multisensory processing [40,80–82]. However, it is important to caution against the 'winner's curse' phenomenon [82], i.e., the tendency for published studies to report inflated effect sizes compared to the actual effect in the population, thereby leading to overly optimistic power calculations for subsequent studies.

Participants were recruited via a psychiatry ward that was specialized for treatment of post-acute psychosis, leading to a nonrepresentative sample of chronic patients with high antipsychotic doses and low psychosis severity (Table 1 and S5 Fig). Healthy participants were included when meeting the inclusion criteria (age 20 to 65, adequate German comprehension, normal or corrected-to-normal vision and audition) and not the exclusion criteria (i.e., no psychiatric or psychosomatic disorders except specific phobias and nicotine addiction; no cardiovascular disorders, diabetes, and neurological disorders). Healthy participants were matched to patients for age, sex, and education (Table 1). Current or former psychiatric disorders were screened using the questions of the structured clinical interview for DSM IV axis I disorders, SCID-I, German version. Patients were included when meeting the same inclusion and exclusion criteria as healthy controls (except psychiatric disorders). To clinically diagnose the patients, they underwent the full SCID-I interview (see below), and 19 of the 26 patients fulfilled the criteria for schizophrenia and were included (12 paranoid, 1 disorganized, 3 undifferentiated, 3 residual). Two of those 19 SCZ patients were excluded because they showed an unusual response profile in the CMB with a very small, outlying task effect (i.e., task effect <2 STD from the mean task effect) suggesting that these 2 patients only used the beeps to provide the auditory and visual numeric reports. Six patients fulfilled the criteria for schizoaffective disorder and 1 patient for substance-induced psychotic disorder and were excluded. Note that we obtained comparable results when we added the 6 schizoaffective patients to the SCZ sample (i.e., yielding n = 23 patients; S1 Text and S8–S11 Figs and S6–S8 Tables).

Of the 17 included SCZ patients in our main report, 4 exhibited comorbid major depression (lifetime), 1 patient alcohol dependence (lifetime), 3 patients cannabis dependence (lifetime, 1 also acute), 1 patient cocaine dependence (lifetime), 1 patient social phobia (lifetime and acute), 2 patients compulsive-obsessive disorder (lifetime), 1 patient binge-eating disorder (acute; for general demographics and other sample characteristics, see Table 1). The mean duration of the psychotic illness in SCZ patients was 13.37 ± 10.75 years (mean ± STD). Fifteen patients were medicated with atypical antipsychotics (amisulpride, aripriprazole, quetiapine, olanzapine, paliperidone), 1 patient was medicated with typical and atypical antipsychotics (haloperidol and clozapine), and 1 patient was unmedicated. Acute positive symptoms were rather moderate in the current sample (cf. PANSS Table 1).

One healthy participant did not attend the interview session and was excluded. One healthy participant reported an asymptomatic arteriovenous malformation. Because behavioral data and EEG were inconspicuous, the participant was included. Overall, 23 healthy controls and 17 SCZ patients were analyzed. Data from the 23 healthy participants were reported in [27].

## Experimental procedures

Participants took part in 2 sessions on 2 separate days. In the first session, participants underwent clinical and neuropsychological assessments (Table 1). For HC, clinical assessments comprised the screening of the SCID-I interview [86]. For SCZ, clinical assessment comprised the

full SCID-I interview, assessment of the Positive and Negative Symptom Scale (PANSS) [87], the German versions of the revised Launay Slade Hallucination Scale (LSHS-R) [88], the Paranoia Checklist with frequency items (PCL) [89], the Calgary Depression Scale for Schizophrenia (CDSS) [90], and a recording of clinical data (e.g., medication). PANSS measures the severity of clinical positive and negative symptoms as well as general psychopathology on seven-point Likert scales in the last 7 days using a semi-structured clinical interview. LSHS-R and PCL questionnaires measure self-reported subclinical and prodromal hallucinatory experiences and paranoid delusional thinking, respectively, on five-point Likert scales, also in nonclinical samples. Distributions of PANSS positive symptoms, LSHS-R and PCL are reported in S5 Fig. The antipsychotic medication was converted to chlorpromazine-equivalent doses [91]. Both groups underwent neuropsychological tests comprising the verbal learning and memory test (VLMT) [92], the Trail Making Test (TMT-A and -B) [93] assessing visual attention and executive functions (i.e., task switching), a test for premorbid crystallized intelligence (Mehrfachwahl-Wortschatz-Intelligenztest, MWT-B) [94], a Stroop test [95] assessing executive functions (i.e., cognitive inhibition), the Edinburgh Handedness Inventory (EHI) [96], and Beck's Depression Inventory (BDI) [97]. All clinical and neuropsychological assessment were made by trained and experienced clinicians. In the second session, participants underwent a flash-beep paradigm including EEG measurements.

## Stimuli

The flash-beep paradigm was an adaptation of the "sound-induced flash illusion" paradigm [57,98]. The visual flash was a circle presented in the center of the screen on a black background (i.e., 100% contrast; Fig 1A) briefly for 1 frame (i.e., 16.7 ms, as defined by the monitor refresh rate of 60 Hz). The maximum grayscale value (i.e., white) of the circle was at radius 4.5˚ with smoothed inner and outer borders by defining the grayscale values of circles of smaller and larger radius by a Gaussian of 0.9˚ STD visual angle. The auditory beep was a pure tone (2,000 Hz; approximately 70 dBSPL) of 27 ms duration including a 10 ms linear on/off ramp. Multiple visual flashes and auditory beeps were presented sequentially at a fixed SOA of 66.6 ms (see below).

## Experimental design

In the flash-beep paradigm, participants were presented with a sequence of (i) 1, 2, 3, or 4 flashes; and (ii) 1, 2, 3, or 4 beeps (Fig 1A). On each trial, the number of flashes and beeps was independently sampled from 1 to 4 leading to 4 levels of numeric audiovisual disparities (i.e., zero = congruent to 4 = maximal level of disparity; Fig 1B). Each flash and/or beep was presented sequentially in fixed temporal slots that started at 0, 66.7, 133, 200 ms. The temporal slots were filled up sequentially. For example, if the number of beeps was 3, they were presented at 0, 66.6, 133, and 200 ms, while the fourth slot was left empty. Hence, if the same number of flashes and beeps was presented on a particular trial, beeps and flashes were presented in synchrony. On numerically disparate trials, the "surplus" beeps (or flashes) were added in the subsequent fixed time slots (e.g., in case of 2 flashes and 3 beeps: we present 2 flash-beeps at 0 and 66.6 ms in synchrony and a single beep at 133 ms).

Across experimental runs, we instructed participants to selectively report either the number of flashes or beeps and to ignore the stimuli in the task-irrelevant modality. Hence, the $4 \times 4 \times 2$ factorial design manipulated (i) the number of visual flashes (i.e., 1, 2, 3, or 4); (ii) the number of auditory beeps (i.e., 1, 2, 3, or 4); and (iii) the task relevance (auditory- versus visual-selective report) yielding 32 conditions in total (Fig 1B). For analyses of the crossmodal bias, we reorganized trials based on their absolute audiovisual numeric disparity ($|n_A-n_V| \in \{1,2,3\}$).

The duration of a flash-beep sequence was determined by the number of sequentially presented flash and/or beep stimuli (see above for the definition of temporal slots). Irrespective of the number of flashes and/or beeps, a response screen was presented 750 ms after the onset of the first flash and beep for a maximum duration of 2.5 s instructing participants to report their perceived number of flashes (or beeps) as accurately as possible by pushing one of 4 buttons. The order of buttons was counterbalanced across runs to decorrelate motor responses from numeric reports. On half of the runs, the buttons from left to right corresponded to 1 to 4 stimuli; on the other half, they corresponded to 4 to 1. After a participant's response, the next trial started after an inter-trial interval of 1 to 1.75 s.

In every experimental run, each of the 16 conditions was presented 10 times. Healthy control participants completed 4 runs of auditory- and 4 runs of visual-selective report in a counterbalanced fashion (except for 1 participant performing 5 runs of auditory and 3 of visual report) and 2 unisensory runs with visual or auditory stimuli only (i.e., 4 unisensory conditions presented 40 times per run). Thus, HC participants completed 10 runs in total. SCZ participants completed 4 to 8 runs of auditory- and visual-selective report, depending on their endurance, and 1 unisensory visual and 1 auditory run. Thus, SCZ participants completed 6 to 10 runs in total (7.47 ± 0.36, mean ± SEM number of runs). To control for time-dependent differences between groups, the trial numbers of behavioral and EEG data for each HC individual were subsampled to match the average trial number of the SCZ group across all analyses (i.e., trials 1–1186 were taken in HC). Before the actual experiment, participants completed 56 practice trials.

## Experimental setup

Psychtoolbox 3.09 [99] (www.psychtoolbox.org) running under MATLAB R2016a (MathWorks) presented audiovisual stimuli and sent trigger pulses to the EEG recording system. Auditory stimuli were presented at ≈ 70 dB SPL via 2 loudspeakers (Logitech Z130) positioned on each side of the monitor. Visual stimuli were presented on an LCD screen with a 60 Hz refresh rate (EIZO FlexScan S2202W). Button presses were recorded using a standard keyboard. Participants were seated in front of the monitor and loudspeakers at a distance of 85 cm in an electrically shielded, sound-attenuated room.

## Overview of GLM and Bayesian modeling analyses for behavioral data

We assessed similarities and differences in perceptual and causal inference between SCZ and HC by combining GLM-based and Bayesian modeling analysis approaches. The GLM-based analysis computed (i) the correlation between true and reported number of flashes/beep for congruent and unisensory trials; and (ii) the CMB which quantified the relative influence of the auditory and the visual numeric stimuli on observers' auditory and visual behavioral numeric reports. The Bayesian modeling analysis fitted BCI models with different decisional strategies and additional heuristic models to the behavioral numeric reports. We then used Bayesian model comparison to determine the model(s) that were the best explanation for the behavioral data in SCZ or HC.

## GLM-based analysis of behavioral data

Our GLM-based analysis of behavioral data focused on response accuracy indices for unisensory and congruent conditions and on the crossmodal bias for incongruent audiovisual conditions. To compare overall task performance between HC and SCZ, we computed participants' response accuracy by correlating participants' numeric report with the true task-relevant number of auditory or visual stimuli (i.e., Pearson correlation coefficient). We then analyzed Fisher

z-transformed correlation coefficients in unisensory and audiovisual congruent conditions using a mixed-model ANOVA with within-participants factors modality (unisensory versus audiovisual) and task relevance (auditory versus visual report) and between-participant factor group (HC versus SCZ) (Fig 1C and Table 2).

We characterized how HC and SCZ participants weight audiovisual signals during multi-sensory perception by computing the CMB from their responses to audiovisual incongruent stimuli [28,29]. The CMB quantifies the relative influence of the auditory ($n_A$) and the visual ($n_V$) numeric stimuli on observers' auditory or visual behavioral numeric reports ($r_{A/V}$):

$$CMB = \frac{r_{A/V} - n_A}{n_V - n_A} \tag{1}$$

To account for response biases, we adjusted $n_A$ and $n_V$ with a linear regression approach across all congruent trials in a participant-specific fashion. In other words, we replaced the true $n_A$ and $n_V$ in the CMB equations with the $n_A$ and $n_V$ predicted based on participants' responses during the congruent conditions [28]. The CMB should be one if observers' reported auditory or visual numbers were solely influenced by the visual number. CMB should be zero if their reported auditory or visual number were solely influenced by the auditory number.

This CMB-based analysis allowed us to compare the audiovisual weight profiles between HC and SCZ using a mixed-model ANOVA with within-participant factors numeric disparity (1, 2, 3) and task relevance (auditory versus visual report) as well as between-participant factor group (HC versus SCZ) (Fig 1D and Table 2). Further, the analysis allowed us to assess whether the weight profiles were qualitatively in line with the principles of Bayesian causal inference. In particular, we would expect a numeric disparity by task-relevance interaction: the difference in crossmodal bias for visual relative auditory report should be greater for large relative to small numeric disparities. Complementary Bayesian ANOVAs were used to provide evidence for no difference between the groups (see section: "Statistical analysis of behavioral data" below).

### Bayesian modeling analysis for behavioral data

The Bayesian modeling analysis fitted several competing computational models to the individual behavioral numeric reports. We then used factorial Bayesian model comparison to determine the model that was the best explanation for observers' behavioral data and compared the results of the model comparison (Fig 3 and S2 Table) and model parameters between HC and SCZ (Fig 4 and Table 3).

In the following, we will briefly describe the competing models which spanned a 5 × 2 factorial model space. The first factor compared 5 model types that differed in their decision function, i.e., how they combined the estimates of the fusion and the segregation components into a final decision. The 3 BCI model combined the fusion and segregation estimates according to the decisional strategies of model averaging, model selection, and probability matching [30] (please see below for further specification). Two additional heuristic models reported the fusion or segregation estimates with a fixed probability (stochastic fusion model) or depending on whether the numeric disparity was greater than a fixed threshold (i.e., fixed-criterion threshold model; [31]). The second factor compared whether auditory and visual variances were equal across all number of beeps/flashes or increased with stimulus number (i.e., scalar-variability model; [52,53]). Details on the BCI model can be found in [23,31]. Specific details on fitting the BCI model to numeric reports in the current experimental paradigm can be found in [27].

Briefly, the generative model of the BCI model (Fig 3A) assumes that common (C = 1) or independent (C = 2) causes are determined by sampling from a binomial distribution with the

causal prior p(C = 1) = p$_{common}$ (i.e., prior "binding tendency") [35]. For a common cause, the "true" number of audiovisual stimuli N$_{AV}$ is drawn from the numeric prior distribution N($\mu_P$, $\sigma_P$). For 2 independent causes, the "true" auditory (N$_A$) and visual (N$_V$) numbers of stimuli are drawn independently from this numeric prior distribution. Sensory noise is introduced by drawing the sensory inputs x$_A$ and x$_V$ independently from normal distributions centered on the true auditory (respectively visual) number of stimuli with parameters $\sigma_A$ (respectively $\sigma_V$). Thus, the basic generative model included the following free parameters: the causal prior p$_{common}$, the numeric prior's mean $\mu_P$ and standard deviation $\sigma_P$, the auditory standard deviation $\sigma_A$, and the visual standard deviation $\sigma_V$. Given the sensory inputs x$_A$ and x$_V$, the observer infers the posterior probability of the underlying causal structure by combining the causal prior with the sensory evidence according to Bayes rule:

$$p(C = 1|x_A, x_V) = \frac{p(x_A, x_V|C = 1)p_{common}}{p(x_A, x_V)} \tag{2}$$

The causal prior quantifies observers' prior belief that flashes and beeps arise from a common cause and should hence be integrated. In the case of a common cause (C = 1), the optimal audiovisual numeric estimate ($\hat{N}_{AV,C=1}$) is obtained by combining the auditory and visual numeric sensory inputs as well as the numeric prior weighted by their relative precisions (i.e., fusion estimate):

$$\hat{N}_{AV,C=1} = \frac{\frac{x_A}{\sigma_A^2} + \frac{x_V}{\sigma_V^2} + \frac{\mu_P}{\sigma_P^2}}{\frac{1}{\sigma_A^2} + \frac{1}{\sigma_V^2} + \frac{1}{\sigma_P^2}} \tag{3}$$

In the case of independent causes (C = 2), the optimal numeric estimates of the unisensory auditory ($\hat{N}_{A,C=2}$) and visual ($\hat{N}_{V,C=2}$) stimuli are independent (i.e., segregation estimate):

$$\hat{N}_{A,C=2} = \frac{\frac{x_A}{\sigma_A^2} + \frac{\mu_P}{\sigma_P^2}}{\frac{1}{\sigma_A^2} + \frac{1}{\sigma_P^2}}, \hat{N}_{V,C=2} = \frac{\frac{x_V}{\sigma_V^2} + \frac{\mu_P}{\sigma_P^2}}{\frac{1}{\sigma_V^2} + \frac{1}{\sigma_P^2}} \tag{4}$$

Crucially, the observers do not know whether signals come from common or independent sources, but needs to infer this from the sensory signals. To account for observers' causal uncertainty, the model computes a final task-relevant numeric estimate ($\hat{N}_A$ or $\hat{N}_V$) by combining the fusion estimate (i.e., $\hat{N}_{AV,C=1}$ formed under C = 1 assumption) and task-relevant segregation estimates (i.e., either $\hat{N}_{V,C=2}$ or $\hat{N}_{A,C=2}$ for C = 2 assumption) depending on the posterior probabilities of the estimates' underlying causal structures (p(C = 1|x$_A$, x$_V$). The observer can combine the numerical estimates according to different decision strategies [30]:

In the "model averaging" strategy, the observers weigh the estimates in proportion to the posterior probabilities of their underlying causal structures:

$$\hat{N}_A = p(C = 1|x_A, x_V)\hat{N}_{AV,C=1} + (1 - p(C = 1|x_A, x_V))\hat{N}_{A,C=2}$$
$$\hat{N}_V = p(C = 1|x_A, x_V)\hat{N}_{AV,C=1} + (1 - p(C = 1|x_A, x_V))\hat{N}_{V,C=2} \tag{5}$$

In the "model selection" strategy, the observer selects the numeric estimate that has a higher posterior probability:

$$\hat{N}_A = \begin{cases} \hat{N}_{A,C=1} & \text{if } p(C = 1|x_A, x_V) > 0.5 \\ \hat{N}_{A,C=2} & \text{if } p(C = 1|x_A, x_V) \leq 0.5 \end{cases}, \hat{N}_V = \begin{cases} \hat{N}_{V,C=1} & \text{if } p(C = 1|x_A, x_V) > 0.5 \\ \hat{N}_{V,C=2} & \text{if } p(C = 1|x_A, x_V) \leq 0.5 \end{cases} \tag{6}$$

In the "probability matching" strategy, the observer selects the numerical estimate stochastically in proportion to the posterior causal probabilities:

$$\hat{N}_A = \begin{cases} \hat{N}_{AV,C=1} \text{ if } p(C=1|x_A,x_V) > \alpha \\ \hat{N}_{A,C=2} \text{ if } p(C=1|x_A,x_V) \leq \alpha \end{cases}, \hat{N}_V = \begin{cases} \hat{N}_{AV,C=1} \text{ if } p(C=1|x_A,x_V) > \alpha \\ \hat{N}_{V,C=2} \text{ if } p(C=1|x_A,x_V) \leq \alpha \end{cases}, \alpha \sim U(0,1) \quad (7)$$

We compared these 3 decision strategies with 2 additional heuristic strategies [31]. In a stochastic-fusion model, observers stochastically choose the audiovisual numeric (i.e., fusion) or the task-relevant unisensory (i.e., segregation) estimate with a fixed probability parameter η:

$$\hat{N}_A = \begin{cases} \hat{N}_{AV,C=1} \text{ if } \eta > \alpha \\ \hat{N}_{A,C=2} \text{ if } \eta \leq \alpha \end{cases}, \hat{N}_V = \begin{cases} \hat{N}_{AV,C=1} \text{ if } \eta > \alpha \\ \hat{N}_{V,C=2} \text{ if } \eta \leq \alpha \end{cases}, \alpha \sim U(0,1) \quad (8)$$

The stochastic-fusion model encompasses a potential "forced" fusion (η = 1) or "forced" segregation (η = 0) as well as any intermediate response behavior (0 < η < 1). Importantly, the stochastic-fusion model responds irrespective of the audiovisual signals' numeric disparity (cf. Fig 1B). Finally, the fixed-criterion model reports the fusion or segregation estimate whenever the signals' absolute numeric disparity is below or above a fixed criterion k. Thus, the fixed-criterion model implements a response heuristic without formally computing the posterior probability of the causal structure as the BCI model:

$$\hat{N}_A = \begin{cases} \hat{N}_{AV,C=1} \text{ if} |x_A - x_V| < k \\ \hat{N}_{A,C=2} \text{ if} |x_A - x_V| \geq k \end{cases}, \hat{N}_V = \begin{cases} \hat{N}_{AV,C=1} \text{ if} |x_A - x_V| < k \\ \hat{N}_{V,C=2} \text{ if} |x_A - x_V| \geq k \end{cases} \quad (9)$$

For the second model factor, we implemented 2 sensory noise models: Sensory standard deviations were either constant across stimulus numbers or increased in proportion to the stimulus controlled by an incremental sensory standard deviation parameter Δσ:

$$\sigma_A' = \sigma_A + \triangle\sigma_A(S_A - 1), \sigma_V' = \sigma_V + \triangle\sigma_V(S_V - 1) \quad (10)$$

This additional increase in standard deviation with stimulus number incorporates the notion of scalar variability. Additionally, all models included a lapse rate parameter L to account for random uniformly distributed responses independent of the audiovisual inputs.

We fitted each of the 10 models in our 5 decision-strategy × 2 sensory-noise factorial model space to the numeric reports individually for each participant (for details on the fitting procedure, see [27]). For more efficient parameter estimation, we fitted models jointly to unisensory and audiovisual conditions. Thus, the auditory/visual standard deviations and their increments, the numeric prior's mean and standard deviation as well as the lapse rate were jointly informed by unisensory and audiovisual conditions.

To obtain maximum likelihood estimates for the 8 parameters of the models ($p_{common}$ / k / η, $\mu_P$, $\sigma_P$, $\sigma_A$, $\sigma_V$, $\Delta\sigma_A$, $\triangle\sigma_V$, L), we used a Bayesian optimization algorithm as implemented in the BADS toolbox [100]. This optimization algorithm was initialized with 50 different random parameters. We report the results (i.e., model comparisons and parameters) for models with the highest log likelihood across these initializations. Parameter recovery showed that the parameters could be reliably identified with very small biases using this fitting procedure (S12 Fig). Further, model recovery [31,101] demonstrated that the BCI and heuristic models were identifiable and our model fitting procedure was reliable (S13 Fig).

To identify the optimal model for explaining participants' data, we compared the 10 candidate models using the Bayesian information criterion (BIC) as an approximation to the model evidence [102]. We performed Bayesian model comparison at the random-effects group level [103], separately for HC and SCZ, as implemented in SPM12 [104]. Thus, we obtained the protected exceedance probability, i.e., the probability that a given model is more likely than any other model, beyond differences due to chance [103], for each of the 10 candidate models (S2 Table). Further, we performed factorial Bayesian model selection on the two-factorial model space using robust Bayesian model-selection procedures (L. Acerbi, personal communication) to compute protected exceedance probabilities for each family across the 2 factors in our $2 \times 5$ factorial model space (Fig 3B). To assess whether the frequencies of the optimal model differ between HC and SCZ, we applied between-group Bayesian model comparison [103] across the 10 candidate models as implemented in the variational Bayesian approach toolbox [105]. Using the posterior probability that the 2 groups have the same model frequencies, we computed the Bayes factor $BF_{10}$ quantifying the evidence in favor of different (H1) or equivalent (H0) model frequencies (i.e., assuming equal prior probabilities).

To investigate whether HC and SCZ differ in parameters of the winning model (i.e., the BCI model with "model averaging" and increasing sensory variances), we compared the parameters between groups using a two-sample two-sided randomization t test ($n$ = 5,000 randomizations; Table 3). Further, we correlated the parameters with SCZ patients' positive and negative symptoms as well as general psychopathy as measured with the PANSS using a randomization test of the correlation ($n$ = 5,000 randomizations).

To investigate whether HC and SCZ differentially adjusted the causal priors depending on the stimulus history, we sorted the current trials according to the previous trials' absolute audiovisual numeric disparity (i.e., $|n_A\text{-}n_V| \in \{0\text{–}3\}$) and selectively refitted the causal prior of the BCI model (Fig 5A). We entered the causal prior $p_{common}$ into a 4 (numeric disparity on previous trial: 0, 1, 2, 3) $\times$ 2 group (HC versus SCZ) mixed-model ANOVA. Likewise, we investigated how observers adapted the numeric prior by sorting the current trials according to the previous trials' task-relevant auditory/visual signals (i.e., $n_A$ or $n_V \in \{1\text{–}4\}$) and selective refitting of the numeric prior (i.e., numeric prior's mean $\mu_P$ and standard deviation $\sigma_P$; Fig 5A and 5B). We entered the numeric prior's mean $\mu_P$ and standard deviation $\sigma_P$ into separate 4 (number of task-relevant signals on previous trial: 1, 2, 3, 4) $\times$ 2 group (HC versus SCZ) mixed-model ANOVAs. To provide evidence for the null-hypothesis, i.e., no difference between groups, these model parameters were also entered into separate Bayesian mixed-model ANOVAs (see statistical analyses below). Parameter recovery showed that the causal and numeric prior could be reliably identified as a function of the previous trials' conditions with very small biases using this fitting procedure (S14 Fig).

To generate predictions for the behavioral CMB index and EEG analyses (see below) based on the BCI model, we simulated new $x_A$ and $x_V$ for 10,000 trials for each of the 32 conditions using the fitted BCI model parameters of each participant (i.e., BCI model with model averaging and increasing sensory variances). For each simulated trial, we computed the BCI model's (i) unisensory visual ($\hat{N}_{V,C=2}$); (ii) unisensory auditory ($\hat{N}_{A,C=2}$) estimates; (iii) forced-fusion ($\hat{N}_{AV,C=1}$); (iv) final BCI audiovisual numeric estimate ($\hat{N}_A$ or $\hat{N}_V$ depending on whether the auditory or visual modality was task relevant). Next, we used the mode of the resulting (kernel-density estimated) distributions for each condition and participant to compute the model predictions for the CMB index (Fig 1D) and decoding the BCI model's estimates from EEG patterns (see multivariate EEG analysis; Fig 6).

## Statistical analyses of behavioral data

To refrain from making parametric assumptions, we compared behavioral measures and parameters (i.e., neuropsychological test scores, CMB and BCI model parameters; Tables 1 and 3) between conditions and groups as well as tested correlations between BCI model parameters and symptom scores using randomization tests with t-values as test statistics ($n$ = 5,000 randomizations). To assess the evidence in favor of the null-hypothesis (i.e., no difference between groups or conditions), we also performed complementary Bayesian analyses. Randomization tests were complemented by Bayes factors $BF_{10}$ quantifying evidence in favor of a condition or group difference or correlation (H1) relative to the null hypothesis of no difference or correlation (H0) using the bayesFactor toolbox [106]. Bayesian $t$ tests and correlations assumed a Jeffrey–Zellner–Siow prior and $t$ tests assumed a scaling factor s = 0.707 of the Cauchy prior on effects [107].

To evaluate whether experimental factors influenced HC and SCZ differentially or equivalently (e.g., for CMB), we complemented classical with Bayesian mixed-model ANOVAs (Tables 2 and 4) with multivariate Cauchy priors on the effects and uniform model priors. Bayesian ANOVAs allowed to compute inclusion Bayes factors ($BF_{incl}$) for each experimental factor which quantify the average evidence that including the factor in the ANOVA models improves the models' fit given a higher complexity of these models [108,109]. For Bayesian analyses, the following interpretations of BF apply: BF > 3 or > 10 provide substantial or strong evidence [110,111] for condition/group differences, correlations or inclusion of a factor, whereas BF < 1/3 or < 1/10 provide substantial or strong evidence for condition/group equivalence, no correlation or exclusion the factor [109]. All classical and Bayesian ANOVAs were computed using JASP 0.17.2.1 [108].

## EEG—Data acquisition and preprocessing

EEG signals were recorded from 64 active electrodes positioned in an extended 10–20 montage using electrode caps (actiCap, Brain Products, Gilching, Germany) and two 32 channel DC amplifiers (BrainAmp, Brain Products). Electrodes were referenced to FCz using AFz as ground during recording. Signals were digitized at 1,000 Hz with a high-pass filter of 0.1 Hz. Electrode impedances were kept below 25 kOhm. Preprocessing of EEG data was performed using Brainstorm 3.462 running on Matlab R2015b. EEG data were band-pass filtered (0.25 to 45 Hz). Eye blinks were automatically detected using data from the FP1 electrode (i.e., a blink was detected if the band-pass (1.5 to 15 Hz) filtered EEG signal exceeded 2 times the STD; the minimum duration between 2 consecutive blinks was 800 ms). Signal-space projectors (SSPs) were created from band-pass filtered (1.5 to 15 Hz) 400 ms segments centered on detected blinks. The first spatial component of the SSPs was then used to correct blink artifacts in continuous EEG data. Further, all data were visually inspected for artifacts from blinks (i.e., residual blink artifacts after correction using SSPs), saccades, motion, electrode drifts, or jumps and contaminated segments were discarded from further analysis. On average, we discarded 6.171 ± 0.923% SEM of all trials for HC and 7.216 ± 1.444% for SCZ; the difference was not significant, $t_{38}$ = −0.638, $p$ = 0.528, d = −0.204, $BF_{10}$ = 0.366. Finally, EEG data were re-referenced to the average of left and right mastoid electrodes and downsampled to 200 Hz. For analysis of ERPs and decoding analyses (see below), all EEG data were baseline corrected with a 200 ms prestimulus baseline and were analyzed from 100 ms before stimulus onset up to 750 ms after stimulus onset, when the response screen was presented.

For multivariate analyses (see below), single-trial EEG data from the 64 electrodes were binned in sliding time windows of 60 ms using an overlap of 40 ms. Hence, given a sampling rate of 200 Hz, each 60 ms time window included 12 temporal sampling points; 64-electrode

EEG activity vectors (for each time sample) were concatenated across the 12 sampling points within each bin resulting in a spatiotemporal EEG activity pattern of 768 features. EEG activity patterns were z scored to control for mean differences between conditions. The first sampling point in the 60 ms time window was taken as the window's time point in all analyses.

## EEG—Analysis of audiovisual interactions in ERPs

In univariate analyses, we assessed whether schizophrenia alters basic sensory components and early audiovisual interactions in occipital ERPs. We averaged trial-wise EEG data time-logged to stimulus onset into ERPs for audiovisual congruent conditions (i.e., controlling for effects of disparity and averaging over effects of task-relevance) and unisensory conditions. We then averaged the ERPs across occipital electrodes (i.e., O1, O2, Oz, PO3, POz, PO4; Fig 7). To analyze early audiovisual interactions, we computed the difference between congruent audiovisual conditions and the corresponding unisensory conditions (i.e., AVcongr–(A + V)). Because attentional and cognitive sets between uni- and bisensory runs might have differed and our experimental design did not include null trials to account for anticipatory effects around stimulus onset [112], the audiovisual interactions need to be interpreted with caution. To test whether the difference waves deviated from zero at the group level across HC and SCZ, we used a nonparametric randomization test (5,000 randomizations) in which we flipped the sign of the individual difference waves using a one-sample $t$ test as a test statistic [113]. To compare the difference waves between both groups, we used a randomization test in which we flipped group membership (5,000 randomizations) using two-sample $t$ tests as a test statistic. To correct for multiple comparisons across the EEG sampling points, we used a cluster-based correction [114] with the sum of the t values across a cluster as cluster-level statistic and an auxiliary cluster-defining threshold of t = 2 for each time point. In a supplementary analysis, we used a group classification approach [72], in which we assessed whether the multivariate EEG response patterns to unisensory and multisensory stimuli differed between HC and SCZ (S1 Text and S7 Fig).

## EEG—Multivariate analyses of BCI estimates

To characterize whether the neurophysiological processes underlying Bayesian causal inference differ between HC and SCZ, we decoded the 4 numerical estimates of the BCI model from multivariate EEG patterns across poststimulus time points using linear support-vector regression [27,36] as implemented in LibSVM 3.20 [115]. The individually fitted BCI model provided the (i) unisensory visual ($\hat{N}_{V,C=2}$); (ii) unisensory auditory ($\hat{N}_{A,C=2}$) estimates; (iii) forced-fusion ($\hat{N}_{AV,C=1}$); (iv) final BCI audiovisual numeric estimate ($\hat{N}_A$ or $\hat{N}_V$) for each of the 32 audiovisual conditions (see above). For each numeric estimate and each time window, an SVR model was trained to decode the numeric estimate from single-trial EEG activity patterns (see above for a definition) across all 32 conditions in a leave-one-run-out cross validation scheme. The SVRs' parameters (C and ν) were optimized using a grid search within each cross-validation fold (i.e., nested cross-validation).

To quantify the decoding accuracies, we computed the Pearson correlation coefficients between the "true" model estimates and the decoded model estimates (Fig 6A). The Fisher's z-transformed correlation coefficients were tested against zero across HC and SCZ using a one-sided randomization test (i.e., sign flip of correlation coefficient in 5,000 randomizations; 1 sample $t$ test as test statistic). The Fisher's z-transformed correlation coefficients were compared between HC and SCZ using a two-sided randomization test (i.e., 5,000 randomizations; two-sample $t$ tests as a test statistic). Further, we applied Bayesian two-sample $t$ tests (as

described above for behavioral data) to compute Bayes factors that quantify evidence for or against group differences in decoding accuracies across poststimulus time points (Fig 6B).

## DRYAD DOI

All raw EEG and behavioral data are available from a Dryad repository [116].

## Supporting information

**S1 Text. Supporting methods and results.**
(DOCX)

**S1 Fig. Numeric reports and predictions from a log-linear regression model as well as the BCI model as a function of the visual signal number ($n_V$), auditory signal number ($n_A$), and task relevance, separately for HC (left 2 columns, *n* = 23) and SCZ (right 2 columns, *n* = 17). (A)** Mean numeric reports (across participants mean ± SEM). **(B)** Mean predictions (across-participants mean) from a log-linear regression model that predicted the numeric reports from the logarithmic visual and auditory signal numbers as well as their interaction (i.e., $r_{A/V} = b_A * \log(n_A) + b_V * \log(n_V) + b_{AxV} * \log(n_V) * \log(n_A) + c$). Statistical results for the regression model can be found in S1 Table. **(C)** Mean predictions from the BCI model (across-participants mean ± SEM; model-averaging with increasing sensory variances) reproduced the logarithmic compression of numeric reports.
(DOCX)

**S2 Fig. Overall reaction time (RT) density in HC and SCZ (*n* = 40).** When fitting ex-Gaussian functions to the RT distributions, the functions' mean ($t_{38} = -1.325$, *p* = 0.193, d = −0.424, $BF_{10}$ = 0.620), variance ($t_{38} = -0.511$, *p* = 0.612, d = −0.164, BF10 = 0.346), and lambda parameter ($t_{38} = -0.607$, *p* = 0.548, d = −0.194, BF10 = 0.361) were not significantly different, but rather equivalent, between both groups.
(DOCX)

**S3 Fig. The strength of the fission and fusion illusions (across-participants mean ± SEM) is shown as a function of group (HC vs. SCZ, *n* = 40).** The illusions were computed as difference in sensitivity (d prime, d') between the unisensory baseline condition (V1A0 vs. V2A0) and the illusion conditions (fission: V1A2 vs. V2A2; fusion: V1A1 vs. V2A2). For the fission or fusion conditions, response "2" or "1," respectively, were defined as signal. Thus, this illusion measure accounts for a possible shift in the response criterion which could be confounded with the fission or fusion illusions (Vanes et al, 2016). In contrast to Vanes and colleagues, a mixed-model ANOVAs did not reveal a significant difference in illusion strength between the 2 types of illusions (factor audiovisual illusion, $F_{1,38}$ = 2.598, *p* = 0.115, part. $\eta^2$ = 0.064, $BF_{incl}$ = 0.461) or a difference between HC and SCZ participants (factor group: $F_{1,38}$ = 1.431, *p* = 0.239, part. $\eta^2$ = 0.036, $BF_{incl}$ = 0.440; interaction illusion × group: $F_{1,38}$ = 1.271, *p* = 0.267, part. $\eta^2$ = 0.032, $BF_{incl}$ = 0.273). Note that we only included trials in which participants counted the number of flashes as $\leq$ 2.
(DOCX)

**S4 Fig. Bayesian information criterion values for each of the 5 models with increasing sensory variance are shown for each of the 17 patients in the SCZ group ranked according to their PANSS positive score as an index of psychosis severity.** Model comparison was computed as BIC of each model relative to the fixed-criterion model with increasing sensory variance, i.e., higher is better. The model comparison included 5 models with increasing sensory variance and the 5 decision strategies (MA, model averaging; MS, model selection; PM,

probability matching; FC, fixed-criterion model; SF, stochastic fusion model). The SCZ patients were rank-ordered according to their PANSS positive score on the x-axis. In cases of ties, patients were arbitrarily rank-ordered within their pair according to participant number. (DOCX)

**S5 Fig. Characterization of psychosis severity. (A)** Data (across-participants mean ± SEM; $n = 17$) of the PANSS scale for positive symptoms (7 items, range 1–7) and the scale average in the SCZ sample. **(B)** Sum score from the LSHS-R questionnaire (12 items, range 0–48) for a subset of the HC sample ($n = 5$) and the SCZ sample ($n = 17$). The difference between both groups is significant (paired $t$ test, $t_{20} = -2.216$, $p = 0.039$, d = −1.127). **(C)** Sum score (across participants mean ± SEM) from the PCL questionnaire (18 items, range 18–90) for a subset of the HC sample ($n = 5$) and the SCZ sample ($n = 16$). The difference between both groups is marginally significant (paired $t$ test, $t_{19} = -2.0271$, $p = 0.057$, d = −1.039). Importantly, both the LSHS-R and the PCL scores varied substantially across the patient group. This allowed us to assess whether these sensitive measures of psychotic symptoms may correlate with BCI model parameters (S3 Table). PANSS: Positive and Negative Symptom Scale; LSHS-R: Launey–Slade hallucination scale; PCL: Paranoia check list. (DOCX)

**S6 Fig. Modulation of crossmodal bias (CMB) and numeric reports by previous audiovisual numeric disparity and task-relevant signal number. (A)** The audiovisual crossmodal bias (CMB; across-participants mean ± SEM; $n = 40$) is shown as a function of a current trial's numeric disparity (1, 2, or 3) × previous trial's numeric disparity (0, 1, 2, 3) × task relevance (auditory vs. visual report) × group (HC vs. SCZ). A four-factorial mixed-model ANOVA including these factors revealed a marginally significant task-relevance × previous-disparity interaction ($F_{2.1,80.9} = 2.616$, $p = 0.076$, part. $\eta^2 = 0.064$; effects of task-relevance, current numeric disparity and their interaction were $p < 0.05$ similarly as reported in Table 2; all other main and interactions effects, especially involving group, $p > 0.05$). Moreover, we assessed the more constrained hypothesis that the difference between CMB for A and V report would increase linearly with increasing numeric disparity of the previous trial: We applied contrast weights to the A-V report differences depending on the disparity at previous trial, i.e., −1.5 for 0, −0.5 for 1, 0.5 for 2, and 1.5 for 3 audiovisual numeric disparity on the previous trial. This linear contrast showed a significant linear modulation of the task relevance effect on the CMB by previous disparity (i.e., linear contrast on the interaction of previous disparity and task relevance; $t_{114} = 2.712$, $p = 0.008$). In other words, audiovisual interactions increased when the previous numeric disparity was small, but decreased when the previous numeric disparity was large. CMB = 1 for purely visual and CMB = 0 for purely auditory influence. **(B)** Visual (left panel) and auditory (right panel) numeric reports (across-participants mean ± SEM, $n = 40$) plotted as a function of current visual or auditory signal number, previous visual or auditory signal number and group (HC vs. SCZ). A three-factorial mixed-model ANOVA including these factors revealed a significant main effect of previous signal number ($F_{1.7,64.7} = 33.192$, $p < 0.001$, part. $\eta^2 = 0.466$), an interaction of previous signal number and task relevance ($F_{3,114} = 3.986$, $p = 0.010$, part. $\eta^2 = 0.095$), and a significant interaction of previous signal number and current signal number ($F_{6.6,250.2} = 8.515$, $p < 0.001$, part. $\eta^2 = 0.183$). The effect of current signal number and the interaction of current signal number and task relevance were also significant ($p < 0.05$). All other main and interactions effects, especially involving group, were not significant. (DOCX)

**S7 Fig. Multivariate ERP patterns, the spatial correlation of ERP patterns between HC and SCZ and decoding results ($n = 40$). (A)** ERP patterns (i.e., topographies) are shown as a

function of time, averaged in 100 ms time windows and separately for HC and SCZ participants. **(B)** Spatial correlation of multivariate ERP patterns between HC and SCZ participants as a function of time, separately for the 4 conditions (N.B.: Spatial correlations before stimulus onset might arise from expectation processes). **(C)** Decoding accuracy (i.e., fraction of correct classifications) of a decoder trained to classify the participant group (i.e., HC vs. SCZ) from ERP patterns of the 4 conditions. No significant clusters ($p > 0.05$) of decoding accuracies above the chance level of 0.5 were found in one-sided cluster-based corrected randomization tests. As a reference, the gray dashed lines indicate decoding accuracy (fraction ± 68% CI; averaged across the 4 conditions) of the decoder trained on randomized group membership ($n = 5,000$ randomizations).
(DOCX)

**S8 Fig. Crossmodal bias for the clinical sample including schizophrenia (SCZ, *n* = 17) and schizoaffective (SCA, *n* = 6) patients.** Participants' audiovisual crossmodal biases (CMB; across-participants mean ± SEM; $n = 46$) are shown as a function of numeric disparity (1, 2, or 3), task relevance (auditory vs. visual report), and group (HC vs. SCZ). CMB = 1 for purely visual and CMB = 0 for purely auditory influence. Source data is provided in S8 Data.
(DOCX)

**S9 Fig. Factorial Bayesian model comparison including SCZ (*n* = 17) and SCA patients (*n* = 6).** Ten models with different decision strategies, model averaging (MA), model selection (MS), probability matching (PM), fixed criterion (FC), stochastic fusion (SF), each with constant or increasing sensory auditory and visual variances. The image shows the relative model evidence for each model (i.e., participant-specific Bayesian information criterion of a model relative to the worst model summed over all participants). A larger model evidence indicates that a model provides a better explanation of our data. Source data is provided in S9 Data.
(DOCX)

**S10 Fig. Parameters of the BCI model (across participants mean ± SEM) separately plotted for HC and SCZ (*n* = 17) & SCA (*n* = 6) patients.** The BCI model's decision strategy applies model averaging with increasing sensory variance. Source data is provided in S10 Data.
(DOCX)

**S11 Fig. Decoding the BCI model's numeric estimates from EEG patterns using support-vector regression (SVR) in HC versus SCZ (*n* = 40) & SCA (*n* = 6) patients. (A)** Decoding accuracy (Fisher's z-transformed correlation; across-participants mean) of the SVR decoders as a function of time and group (HC vs. SCZ & SCA). Decoding accuracy was computed as Pearson correlation coefficient between the given BCI model's internal estimates and BCI estimates that were decoded from EEG activity patterns using SVR models trained separately for each numeric estimate. Color-coded horizontal solid lines (HC) or dashed lines (SCZ & SCA) indicate clusters of significant decoding accuracy ($p < 0.05$; one-sided one-sample cluster-based corrected randomization *t* test). **(B)** Bayes factors for the comparison between the decoding accuracies of HC and SCZ & SCA for each of the BCI estimate (i.e., BF10 > 3 substantial evidence for or BF10 < 1/3 against group differences). Source data is provided in S11 Data.
(DOCX)

**S12 Fig. Results of the parameter recovery (*n* = 40).** For parameter recovery, the winning model (i.e., model averaging with increasing sensory variances) predicted responses which were then again fitted to obtain recovered parameters with the same fitting procedure as for the main analysis (i.e., initialization with 50 different random parameters; predicted

distributions were generated from 5,000 simulated trials per condition). The plots show the recovered parameters as a function of the parameters originally fitted to participants' behavioral data. The red line is a line with slope 1 and intercept 0.
(DOCX)

**S13 Fig. Results of model recovery.** To assess model recovery, each of the 5 models with different decision strategies (model averaging, MA; model selection, MS; probability matching, PM; fixed criterion, FC; stochastic fusion, SF) and increasing sensory variance parameters generated responses in 20 randomly selected participants. Each of the 5 models was then fitted to the generated responses of each model with the same fitting procedure as for the main analysis (i.e., initialization with 50 different random parameters; predicted distributions were generated from 5,000 simulated trials per condition). The confusion matrix shows for each generating model (rows) the fraction of participants in which a fitted model (columns) won the model comparison across the 5 fitted models based on Bayesian information criterion within an individual. The bright diagonal indicates that the generating models won the model comparisons, indicating successful model recovery. In particular, the winning model with model averaging as decision strategy was correctly recovered in 85% of the cases, thereby supporting the validity of our model-based analyses.
(DOCX)

**S14 Fig. Results of the parameter recovery (HC and SCZ, *n* = 40) for the modeling analysis of updated causal and numeric priors.** For parameter recovery, the model averaging model with updated priors predicted responses which were then again fitted to obtain recovered parameters with the same fitting procedure as for the main analysis (i.e., initialization with 50 different random parameters; predicted distributions were generated from 5,000 simulated trials per condition). The plots show the recovered parameters as a function of the parameters originally fitted to participants' behavioral data. Parameters are pooled across different levels of previous trial conditions. The red line is a line with slope 1 and intercept 0.
(DOCX)

**S1 Table. Statistical significance of regression parameter estimates across and between HC and SCZ in the log-linear regression model that predicted the numeric reports from the visual and auditory signal numbers as well as their interaction.**
(DOCX)

**S2 Table. Results of the Bayesian model comparison of the 2 × 5 factorial model space with factor "decision strategy" and "sensory variance" in HC and SCZ participants.**
(DOCX)

**S3 Table. Pearson correlations of BCI model parameters with SCZ patients' (*n* = 17) positive and negative symptoms measured by PANSS, LSHS-R, and PCL.**
(DOCX)

**S4 Table. Results of testing the decoding accuracies of the BCI estimates from EEG patterns against zero in HC, SCZ, and their group difference.**
(DOCX)

**S5 Table. Spearman rank correlations of BCI model parameters with SCZ patients' neurocognitive test scores (*n* = 17).**
(DOCX)

**S6 Table. Main and interaction effects of task relevance (TR), disparity (Disp), and group (HC, *n* = 23 vs. schizophrenia, *n* = 17, and schizoaffective patients, *n* = 6) on the**

**crossmodal bias from classical and Bayesian mixed-model ANOVAs.**
(DOCX)

**S7 Table. Comparison of BCI model parameters between HC ($n$ = 23) versus schizophrenia ($n$ = 17) & schizoaffective ($n$ = 6) patients.**
(DOCX)

**S8 Table. Correlations of BCI model parameters with patients' (schizophrenia, $n$ = 17 and schizoaffective, $n$ = 6) positive and negative symptoms measured by PANSS, LSHS-R, and PCL.**
(DOCX)

**S1 Data. Source data for Fig 1 which can be opened with Matlab or GNU Octave.**
(ZIP)

**S2 Data. Source data for Fig 2 which can be opened with Matlab or GNU Octave.**
(ZIP)

**S3 Data. Source data for Fig 3 which can be opened with Matlab or GNU Octave.**
(ZIP)

**S4 Data. Source data for Fig 4 which can be opened with Matlab or GNU Octave.**
(ZIP)

**S5 Data. Source data for Fig 5 which can be opened with Matlab or GNU Octave.**
(ZIP)

**S6 Data. Source data for Fig 6 which can be opened with Matlab or GNU Octave.**
(ZIP)

**S7 Data. Source data for Fig 7 which can be opened with Matlab or GNU Octave.**
(ZIP)

**S8 Data. Source data for S8 Fig which can be opened with Matlab or GNU Octave.**
(ZIP)

**S9 Data. Source data for S9 Fig which can be opened with Matlab or GNU Octave.**
(ZIP)

**S10 Data. Source data for S10 Fig which can be opened with Matlab or GNU Octave.**
(ZIP)

**S11 Data. Source data for S11 Fig which can be opened with Matlab or GNU Octave.**
(ZIP)

## Acknowledgments

We thank Luigi Acerbi for sharing code to perform robust Bayesian model-selection procedures which was updated by David Meijer. We thank Ramona Täglich and Larissa Metzler for help with the data collection.

## Author Contributions

**Conceptualization:** Tim Rohe, Klaus Hesse, Ann-Christine Ehlis.

**Data curation:** Tim Rohe.

**Formal analysis:** Tim Rohe.

**Funding acquisition:** Tim Rohe.

**Investigation:** Tim Rohe, Klaus Hesse.

**Methodology:** Tim Rohe, Uta Noppeney.

**Supervision:** Ann-Christine Ehlis, Uta Noppeney.

**Writing – original draft:** Tim Rohe, Uta Noppeney.

**Writing – review & editing:** Tim Rohe, Klaus Hesse, Ann-Christine Ehlis, Uta Noppeney.

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
