## [Editor Report · Decision Letter 0]

1 Sep 2023

Dear Dr Rohe, 

Thank you for submitting your manuscript entitled "Computations and neural dynamics of audiovisual causal and perceptual inference in schizophrenia" for consideration as a Update Article by PLOS Biology.

Your manuscript has now been evaluated by the PLOS Biology editorial staff and I am writing to let you know that we would like to send your submission out for external peer review. However, please note that we have yet to make a firm call about whether the study offers a sufficient conceptual advance for PLOS Biology and that we will be looking for strong support from the reviewers.

Once your full submission is complete, your paper will undergo a series of checks in preparation for peer review. After your manuscript has passed the checks it will be sent out for review. To provide the metadata for your submission, please Login to Editorial Manager (https://www.editorialmanager.com/pbiology) within two working days, i.e. by Sep 03 2023 11:59PM.

Kind regards,

Christian

Christian Schnell, PhD

Senior Editor

PLOS Biology

cschnell@plos.org

---

## [Decision Letter · Decision Letter 1]

1 Nov 2023

Dear Dr Rohe,

Thank you for your patience while your manuscript "Computations and neural dynamics of audiovisual causal and perceptual inference in schizophrenia" was peer-reviewed at PLOS Biology. Please allow me to apologize for the long delay in sending our decision, which was caused by difficulties in finding suitable reviewers and academic editors for your study. In any case, your manuscript has now been evaluated by the PLOS Biology editors, an Academic Editor with relevant expertise, and by several independent reviewers. 

In light of the reviews, which you will find at the end of this email, we would like to invite you to revise the work to thoroughly address the reviewers' reports.

As you will see below, the reviewers say that the study is overall well executed and interesting. However, all reviewers raise concerns about the small sample size, concerns about the methodological approach, and mention that the high-level claims are too strong considering the limitations of the work. Apart from the limitation of the small sample size (which should be clearly discussed), it seems that most of the concerns can be addressed with additional analyses and a more careful presentation and discussion of the previous literature and results of the current manuscript. We also encourage you to try to make your paper as accessible as possible to readers who are not familiar with the Bayesian modeling procedures. 

Given the extent of revision needed, we cannot make a decision about publication until we have seen the revised manuscript and your response to the reviewers' comments. Your revised manuscript is likely to be sent for further evaluation by all or a subset of the reviewers.

**IMPORTANT - SUBMITTING YOUR REVISION**

*Re-submission Checklist*

*Published Peer Review*

*PLOS Data Policy*

*Blot and Gel Data Policy*

Sincerely,

Christian

Christian Schnell, PhD

Senior Editor

PLOS Biology

cschnell@plos.org

REVIEWS:

Reviewer's Responses to Questions

Do you want your identity to be public for this peer review?

Reviewer #1: No

Reviewer #2: Yes: Guillermo Horga, MD PhD

Reviewer #3: No

Reviewer #1: This article investigates audiovisual perception in a sound-induced flash illusion paradigm using behavioral, computational (Bayesian modelling) and neuroimaging analysis in 17 SCZ patients as compared to 23 controls. The results are essentially negative: the authors found comparable behavioural, computational and neural mechanisms/ EEG signatures in SCZ and HC an no difference in the model parameters or winning model. Bayesian inference and more particularly causal inference has recently been proposed to be at the origin of differences in SCZ (as well as ASD) so the line of work is important. 

Though dense, the article is very clearly written. It is using a 1-4 beeps x 1-4 flash factorial experimental design which is interesting, the computational analysis is thorough and the EEG decoding exercise interesting. 

In my understanding (apologies if I have missed relevant information), the paper seems to suffer from a few limitations however, which may limit its impact and significance:

- One obvious issue with the study is the sample size. 17 SCZ is a small sample. I couldn't find a power analysis to motivate the sample size of 17 (some indication for supporting a somewhat larger sample size of the 26 initial participants). As is common for such clinical studies, the SCZ group is also very heterogeneous both in terms of comorbidities and treatment. The SCZ group has been medicated for a long time and stable with low PANSS positive scores. It is also apparent in the modelling that the individual parameters are very widespread.  

- It is also unclear whether the experimental design is well chosen to identify differences in SCZ. The SOA is not varied (and the choice of SOA is not very clear motivated either) which could perhaps explain the negative results, in particular if the differences are about the temporal binding window. I think better motivation with respect to previous literature and experimental design and maybe replication of conditions where differences in schizophrenia were found would have helped. 

- For the modelling part, Figure S5 seems to be showing parameter recovery for the winning model rather than model recovery (as indicated in the caption?). It is unclear then what the performance is in terms of model recovery? Similarly, it is not very clear whether the winning is actually a good model for the data (Wilson & Collins, 2019).

- Some aspects are unclear in the text, e.g. are the behavioural effects that are significant (numeric prior and visual variance) corrected for multiple comparisons? 

- Page 12: is the SVM trained on both HC and SCZ data, together or separately? The hypotheses being tested here could be clearer. 

- "In both HC and SCZ, the decoders predicted the BCI estimates from EEG patterns significantly better than chance over most periods of the post-stimulus time window" "Significantly better than chance" is vague, it is frustrating to have to go to the figure to have an idea how well the decoding is doing and it is unclear what level of accuracy would be necessary for making clear conclusions. 

Reviewer #2: Rohe et al. use an EEG-compatible multimodal sensory-integration task to evaluate causal Bayesian inference in medicated patients with schizophrenia and healthy controls. The authors then use computational modeling and EEG decoding to characterize behavioral and neural computations underlying various levels of inference. Overall, they conclude that neither behavioral nor neural data suggest alterations in inference are present in medicated patients with schizophrenia. This article presents a compelling experimental framework and several interesting analyses. Overall, the experimental design and analytical techniques are of excellent quality, particularly for a clinical study. 

Despite my general enthusiasm, however, I have concerns about the characterization of prior literature, interpretation of the results, and some analysis decisions that should be addressed before I can make a recommendation for publication. 

In general terms, and despite the obvious quality of this work, it is not clear what the main takeaways and contributions to this literature are. Given the sample characteristics (see below), I believe it is fair to say that this study is not well positioned to rule in/out prior over/underweighting as a model of psychotic states, and therefore it cannot resolve mixed findings in this literature. The paper's contributions should be made clearer (novel methods and proof-of-concept application to this area?) and the conclusions should be very explicitly qualified with these limitations in mind (particularly in the abstract and the discussion sections, and with particular attention to the strength of evidence for the null hypothesis and generalizability of these conclusions). 

Major concerns

1. Throughout, the authors treat schizophrenia and psychosis interchangeably, from presentation of the literature to analytical justification. This is problematic. Strong/precise prior models (e.g., Powers et al., 2017) are models of psychosis severity and not models of schizophrenia. Importantly these models assume state-dependence of neurocognitive phenotypes of psychosis (Kafadar et al., 2022). So it is critical to separate state-dependent effects from trait-like effects in schizophrenia patients that could be linked to negative or cognitive symptoms, and which have been associated with distinct, even opposite mechanisms from those associated selectively with positive symptoms (e.g., see work from Rick Adams). This confusion arises throughout the manuscript (specific comments below).

a. Description of previous literature: The discussion of the link between hallucinations and prior weighting (citations 8-17) does not distinguish between findings that compare patients vs. controls and studies that consider psychosis (or psychosis-like) severity. Overall the balance of the literature presented gives the wrong impression. The majority of the literature points to a positive relationship between psychosis severity and prior weighting, even if a minority of studies obtained conflicting findings. The literature review should be balanced in terms of the evidence for prior overweighting in patients with more severe psychotic symptoms (hallucinations more concretely) vs. evidence against this proposition. Again, this should separate psychotic states/severity versus trait-like schizophrenia phenotypes (the latter of which, but not the former, are associated with cognitive deficits and amotivation, which can broadly affect cognitive performance). There should also be a distinction between clinical and non-clinical populations.

b. Data analysis: In general, patient vs control comparisons are tested when analyses across psychosis severity (with particular emphasis on hallucination severity) would also be appropriate. There should also be a clearer justification for the scales and items used in the correlations with psychosis and there should be a clearer description of the distribution of psychosis severity in this sample, with emphasis on whether this small sample has enough variability to really test psychosis-related models which require sufficient range of psychosis severity.

c. Results interpretation: This data doesn't appear to argue against strong prior models of psychosis but rather suggests that inference processes are largely spared in a population of medicated schizophrenia patients with low psychosis severity. To the contrary, the authors do report effects of prior variance that relate to psychosis, consistent with many findings in the literature. 

i. In the discussion, the authors note that "we cannot exclude the possibility that SCZ may have been less vigilant … which could be misinterpreted as overreliance on their numeric prior." Wouldn't the authors suspect that individual differences in vigilance would be captured by the lapse term? Which would be consistent with the finding that this term correlates with negative symptoms? To clarify the specificity of the identified relationships between prior variance and positive symptoms, negative and general symptoms should be statistically controlled for in these analyses.

2. Sample: The sample is relatively small which should be acknowledged when arguing for the null hypothesis. The sample is also possibly not representative of the population of patients with schizophrenia as a whole, particularly in early or untreated states. The patients have relatively high doses of antipsychotics, low psychosis severity, and are largely cognitively intact. Furthermore, the recruitment strategy/recruitment sources is not discussed, which may be relevant to characterizing this cohort. Some of these points are acknowledged in the discussion but should be clarified during key takeaway statements throughout the manuscript. This is especially important since the authors seem to be arguing that the results are contradictory with the rest of the literature. 

a. Schizoaffective patients were excluded in this analysis which is very uncommon in this literature. Given that including them would substantially increase the sample size I would like to see analyses including these patients.

3. The authors evaluate evidence for trial-by-trial learning by splitting the data into subsets with different trial histories and fitting the BCI model to these subsets. This approach is not standard and should be more carefully validated and supported. The authors should first demonstrate evidence for these effects using model-agnostic approaches (e.g., lagged-regression predicting report as a function of previous trial identity), showing that their model-based results capture qualitative features of the raw data. As it stands, it is unclear whether there are even meaningful trial-history effects on this task, or whether they should be expected based on this literature. Second, they should demonstrate that this fitting approach produces valid and interpretable parameter estimates via a parameter recovery. Alternatively, there are models in the literature that dynamically update priors as a function of trial-by-trial evidence. A model with this structure would be a more appropriate test of this question. That said, an alternative would be to eliminate this section altogether, since I am not convinced it adds substantial value if the obtained parameters too noisy to provide meaningful information (particularly meaningful null findings) in this small sample.

Minor concerns

1. Most psychotic symptoms are not multisensory even though they can manifest in different modalities (e.g., even in patients who report visual and auditory hallucinations, integrated audiovisual hallucinations are rare). The focus on multisensory perception/integration should be more carefully justified. One clear strength is the ability to investigate different priors and hierarchical levels of inference. This is linked to point 1 above; cited studies (citations 30-36) focus on diagnosis comparisons rather than psychosis comparisons although the text does not make this distinction.

2. Average fit quality by group is shown. Is there substantial variability in the winning model by individual? If so, it would be important to show the proportions of winning models across groups and by psychosis severity.

3. Comparisons of model fitted variances (visual, auditory, and prior) with specific symptoms of psychosis (hallucinations vs delusions) should be shown.

Reviewer #3: Strengths of this paper are that it generally seems like a good next step in understanding whether and how integration of prior information and sensory inputs may be impacted in individuals with schizophrenia. The methods are sound, including useful details about the paradigm, analyses methods and statistics. It is also fairly clearly written and results interpreted reasonably. The study is novel in its use of psychophysical and EEG data as well as computational modeling to answer some research questions regarding causal inference. One major weakness is the small sample size. The findings supported some of the predictions, and these are reasonably well addressed and seem interesting. They stand to add somewhat to the body of knowledge advancing how we think about multisensory integration mechanisms from a Bayesian standpoint. However, not all predictions panned out, reducing impact of the paper. Further, while at a theoretical level, the explanation for the modeling is clear—how each parameter indexes decision processes needs to be clarified further. My comments/ queries are aimed at improving the clarity of the findings and placing within the broader context of clinical research as well.

1) It would be useful to briefly explain more about the modeling, especially the fused weighting, and setting up/ providing detailed rationale for the computational aspect in the introduction

2) ? It is mentioned that "the visual variance" correlated with positive symptoms--- This should be explained more in terms of which exact measure and what this correlation signifies in the context of overweighting priors. Were there any other symptom correlates in the SZ group. Any effect of MWT-B or VLMT scores?

3) While the paradigm, analyses and results attempt to answer the question of sensory integration in terms of stimulus complexity , it would be useful to place these results in the context of other findings on overly strong priors e.g. Pavlovian conditioning and failures in perceptual updating and discussion on what implication the intricacies of multisensory integration have on specific symptoms—This would strengthen the discussion on specificity and clinical implications of this work 

4) Limitations of the sample size and impact of the results/interpretability should be discussed

---

## [Decision Letter · Decision Letter 2]

17 Jul 2024

Dear Dr Rohe,

Thank you for your patience while we considered your revised manuscript "Computations and neural dynamics of audiovisual causal and perceptual inference in schizophrenia" for publication as a Update Article at PLOS Biology. This revised version of your manuscript has been evaluated by the PLOS Biology editors, the Academic Editor and the original reviewers.

Based on the reviews and on our Academic Editor's assessment of your revision, we are likely to accept this manuscript for publication, provided you satisfactorily address the remaining points raised by the reviewers. Please also make sure to address the following data and other policy-related requests.

* We would like to suggest a different title to improve accessibility: "No evidence for impaired multisensory perceptual and causal inference in medicated post-acute individuals with schizophrenia"

* Please provide information in the methods section that the study has been conducted according to the principles expressed in the Declaration of Helsinki.

* Please also attend to the following points: 

** Page 9 line 15, missing the word “bias”

** Page 15 line 34 “accuracyin”

** Fig 1C, the y-axis label “correlation” is not explained or described it seems.

* Please include qualifiers where necessary to indicate that due to the small sample size, the evidence for a certain claim may be limited. Reviewer 1 mentions one of these statements in the abstract where this will be needed. Papers such as Gilles E. Gignac ⁎, Eva T. Szodorai. Effect size guidelines for individual differences researchers, Personality and individual differences (2016) and Button, K. S., Ioannidis, J. P., Mokrysz, C., Nosek, B. A., Flint, J., Robinson, E. S., & Munafo, M. R. (2013) might be useful for further discussing this issue, also considering that your power calculation is based on a large effect size (d=1) which may not be realistic given that effect sizes are usually rather small in this field (e.g., d=0.4).

* DATA POLICY:

Regardless of the method selected, please ensure that you provide the individual numerical values that underlie the summary data displayed in the following figure panels as they are essential for readers to assess your analysis and to reproduce it: 1CD, 4, 5ABC and similar panels in the supplementary information.

* CODE POLICY

We expect to receive your revised manuscript within two weeks. 

*Published Peer Review History*

*Press*

Sincerely,

Christian

Christian Schnell, PhD

Senior Editor

cschnell@plos.org

PLOS Biology

Reviewer remarks:

Reviewer #1: The authors have kindly answered our questions, clarified their methods and discussed the limitations of the study, and updated the manuscript accordingly. 

Some aspects of the work are interesting and solid. 

However, the limitations remain, particularly the fact that the authors mostly report negative findings with a very small sample (which is not well justified by a power analysis, and is poorly representative of a population of patients with schizophrenia as a whole, particularly in early or untreated states), using a design that had not been used before to show differences in SCZ (so not easily comparable to previous literature). The results concerning the numeric prior (the significance of which, unlike that of the p_common prior, was a bit unclear to me to start with) are now muddled/weaker (not significant) after inclusion of the 6 patients with schizo-affective disorder. 

All in all, although I recognise how hard it is to get bigger samples of patients with schizohprenia and/or collecting data on different conditions (e.g. also varying SOAs to explore temporal windows), my opinion is that the contribution and significance of the presented results are limited (if the aim is to present them to a large/general audience). I don't think it is fair for example to express in such broad terms that the results " demonstrate that the neurocomputational mechanisms of multisensory perceptual and causal inference remain remarkably intact in medicated post-acute individuals with schizophrenia" (as written in the abstract) as the results may only hold for the chosen design (which doesn't explore differences in binding windows / replicate situations where differences have been observed) and may miss subtle differences that would be detected in a larger sample. 

Reviewer #2 (Guillermo Horga): The authors have been very responsive and addressed all my comments. This is a great paper and an important contribution to the literature.

---

## [Editor Report · Decision Letter 3]

6 Aug 2024

Dear Tim,

Thank you for the submission of your revised Update Article "Multisensory perceptual and causal inference is largely preserved in medicated post-acute individuals with schizophrenia" for publication in PLOS Biology. On behalf of my colleagues and the Academic Editor, Simon van Gaal, I am pleased to say that we can in principle accept your manuscript for publication, provided you address any remaining formatting and reporting issues. These will be detailed in an email you should receive within 2-3 business days from our colleagues in the journal operations team; no action is required from you until then. Please note that we will not be able to formally accept your manuscript and schedule it for publication until you have completed any requested changes.

PRESS

Sincerely, 

Christian

Christian Schnell, PhD

Senior Editor

PLOS Biology

cschnell@plos.org